# DATA SHAPLEY IN ONE TRAINING RUN

**Jiachen T. Wang**
Princeton University

**Prateek Mittal**
Princeton University

**Dawn Song**
UC Berkeley

**Ruoxi Jia**
Virginia Tech

## ABSTRACT

Data Shapley offers a principled framework for attributing the contribution of data within machine learning contexts. However, the traditional notion of Data Shapley requires re-training models on various data subsets, which becomes computationally infeasible for large-scale models. Additionally, this retraining-based definition cannot evaluate the contribution of data for a specific model training run, which may often be of interest in practice. This paper introduces a novel concept, *In-Run Data Shapley*, which eliminates the need for model retraining and is specifically designed for assessing data contribution for a particular model of interest. In-Run Data Shapley calculates the Shapley value for each gradient update iteration and accumulates these values throughout the training process. We present several techniques that allow the efficient scaling of In-Run Data Shapley to the size of foundation models. In its most optimized implementation, our method adds negligible runtime overhead compared to standard model training. This dramatic efficiency improvement makes it possible to perform data attribution for the foundation model pretraining stage. We present several case studies that offer fresh insights into pretraining data's contribution and discuss their implications for copyright in generative AI and pretraining data curation.

## 1 INTRODUCTION

In today's data-driven world, understanding the contribution of each data point is crucial, especially with the advent of foundation models that rely on vast amounts of training data from various sources. The lack of reliable data attribution mechanisms can lead to significant legal and societal issues, resulting in a growing backlash against the broader use of data for model training (Heikkilä, 2023). For instance, there is a risk of violating intellectual property rights, failing to fairly compensate data creators, and disincentivizing them from producing new, high-quality content (Henderson et al., 2023). This has already resulted in legal disputes, such as the New York Times' lawsuit against Microsoft/OpenAI (Grynbaum & Mac, 2023). Moreover, foundation models are often trained on massive datasets scraped from the internet, which can include low-quality and harmful content (Gao et al., 2020; Raffel et al., 2020; Touvron et al., 2023). Problematic data not only wastes computational resources but also skews model outputs, potentially leading to biased or inaccurate results. By understanding the contribution of each data source, we can identify and mitigate the influence of low-quality data, thereby improving the efficiency and quality of model training.

Since the training data of foundation models comes from multiple stakeholders, it is essential to have an algorithm that can *fairly* attribute data contributions. In recent years, significant progress has been made in understanding what it means to fairly quantify and attribute data source contributions, with *the Shapley value* (Shapley, 1953) emerging as a widely adopted framework (Ghorbani & Zou, 2019; Jia et al., 2019b). Originating from cooperative game theory, the Shapley value uniquely satisfies several desirable properties: (1) It assigns equal scores to equally impactful data points, ensuring fairness; (2) the sum of contribution scores equals the total utility, meaning the scores always represent a share of the total utility; (3) it supports additive decomposition across multiple utility functions, allowing for the calculation of contributions to the entire test set by summing the scores of individual test points. As the Shapley value uniquely satisfies these properties, it eliminates the uncertainty and ambiguity surrounding which attribution frameworks should be used conceptually. While there are other non-Shapley data attribution frameworks, such as Datamodels (Park et al., 2023; Ilyas et al., 2022) and influence functions (Koh & Liang, 2017), they lack the clear theoretical foundation and uniqueness provided by the Shapley value.

**Original Data Shapley definition faces computational & conceptual limitation.** However, Data Shapley, i.e., the application of the Shapley value in data attribution, has been limited to very small-scale models. Existing methods (Ghorbani & Zou, 2019) to estimate the Shapley value require retraining the model numerous times using different subsets of data to evaluate the contribution of each data source, making them computationally infeasible for foundation models. On the other hand, retraining-based methods suffer from a conceptual issue often overlooked in the literature: **they assess data contribution for a general learning algorithm rather than a particular model.** While one might interpret the former as an approximation of the latter, these two quantities can be quite different in practice, especially when the learning algorithm is randomized and sensitive to factors like random initialization and training data order. In many real-life scenarios, however, the primary interest lies in understanding data contribution to the specific model being trained and deployed.

This paper introduces *In-Run Data Shapley*, a novel approach that makes fair data attribution applicable to large-scale foundation models. Unlike retraining-based Data Shapley, In-Run Data Shapley quantifies the contribution of each data source to *the specific target model of interest*.

**Technical contributions.** Our key insight is that ML models are trained using iterative algorithms, where the model performance change in one iteration is sufficiently small to be accurately approximated by first- or second-order Taylor expansions. We show that the Shapley value for the approximated one-step model performance change can be derived analytically via gradient dot-products or gradient-Hessian-gradient products between training and validation data. Hence, we can compute the Data Shapley scores for each model update step, and accumulate the scores throughout the training process. However, the per-sample gradient vectors required for computing the Shapley value in each training iteration introduce significant overhead due to per-sample gradient calculation. To address this challenge, we develop a series of technical tools that enable the exact calculation of gradient dot-products and gradient-Hessian-gradient products in one and two backward passes, respectively, without the need to instantiate any additional gradient vectors or Hessian matrices. Collectively, these tools allow for the efficient computation of In-Run Data Shapley. In particular, with sufficient GPU memory, its most efficient implementation is as fast as regular training.

**Empirical implications.** Given the efficient algorithms developed in this paper, for the first time, one can perform data attribution on the scale of foundation model pretraining. While in this paper, we focus on GPT2 and Pythia-410M as a pilot study, our approach is applicable to larger-scale industrial models with sufficient computing resources. We performed various case studies that provide fresh insights into training data's contribution to the foundation model pretraining.

**(1) There is considerable room for improvement in data curation for pretraining (Section 5.3.1).** Even well-curated pretraining corpora contain data points that negatively impact the training process. We demonstrate the effectiveness of In-Run Data Shapley in identifying these low-quality data points. By computing In-Run Data Shapley values during training and removing negatively valued data points, we show that the cleaned dataset leads to significantly faster model convergence and improved performance compared to the original dataset. Interestingly, despite the Pile dataset (Gao et al., 2020) already undergoing multiple layers of curation, In-Run Data Shapley assigns negative values to approximately 16% of the data. We found a significant amount of noisy data among them, highlighting the need for improved data curation for foundation model training.

**(2) Data's contribution is stage-dependent (Section 5.3.2).** In-Run Data Shapley can capture the dynamics of contribution through the course of training, a fine-grained aspect that cannot be captured by prior works. In-Run Data Shapley shows that in the early stages of training, general corpora tend to have a relatively large contribution regardless of the downstream tasks. This is because general corpora help the model learn basic language patterns, grammar, and common knowledge. However, in the later stages of training, the contribution from domain-specific corpora becomes dominant, and the contribution of the general corpus phases out.

**(3) Rethinking copyright in generative AI: contribution beyond memorization (Section 5.3.3).** We studied training data's contribution to validation points of varying similarity levels. We found that even when the validation data is a complete rewrite of the training data while maintaining the topic, the training data still contributes significantly. This finding has implications for the current dialogue around what constitutes a copyright violation in generative AI (Mulligan & Li, 2024). While the unfair use of copyrighted content is generally only considered when the generated data is an almost verbatim replication of the training data, our contribution analysis shows that some data owners should receive a certain royalty share for generated content, even if the output does not closely resemble the copyrighted material.

## 2 BACKGROUND OF DATA SHAPLEY

In this section, we formalize the setup of data attribution for ML and revisit Data Shapley's definition.

**Setup & Goal.** Given a dataset $\mathcal{D}_{\text{tr}} := \{z_i\}_{i=1}^N$, data attribution or valuation aims to assign a score to each training data point $z_i$, reflecting its importance for the trained ML model's performance on a certain task. Formally, we seek a score vector $(\phi_{z_i})_{i=1}^N$ where each $\phi_{z_i} \in \mathbb{R}$ reflects the *value* of $z_i$.

The Shapley value (SV) (Shapley, 1953), originating from game theory, stands out as a distinguished method for equitably distributing total profit among all participating players. Before diving into its definition, we first discuss a fundamental concept: the *utility function*.

**Utility function.** A *utility function* maps an input dataset to a score indicating the utility of the dataset for model training. In most of the existing literature (Ghorbani & Zou, 2019; Jia et al., 2019b), the utility function $U$ is chosen as the performance (e.g., accuracy or loss) of the trained models on a hold-out validation set. That is, given a training set $S$, the utility function $U(S) := \texttt{Perf}(\mathcal{A}(S))$, where $\mathcal{A}$ represents a learning algorithm that trains a model on dataset $S$, and $\texttt{Perf}(\cdot)$ is a function assessing the model's performance. For example, $\texttt{Perf}(\cdot)$ can be the accuracy for a classification task or the perplexity for a language completion task, evaluated on a (set of) hold-out validation data.

**Definition 1** (Shapley value (Shapley, 1953)). *Let $U(\cdot)$ denote a utility function and $D$ represent a training set of $N$ data points. The Shapley value, $\phi_z(U)$, assigned to a data point $z \in D$ is defined as $\phi_z(U) := \frac{1}{N} \sum_{k=1}^N \binom{N-1}{k-1}^{-1} \sum_{S \subseteq D_{-z}, |S|=k-1} [U(S \cup \{z\}) - U(S)]$ where $D_{-z} = D \setminus \{z\}$.*

In simple terms, the Shapley value is a weighted average of the *marginal contribution* $U(S \cup \{z\}) - U(S)$, i.e., the utility change when the point $z$ is added to different $S$s. For simplicity, we often write $\phi_z$ when the utility function is clear from the context. The popularity of the Shapley value is attributable to the fact that it is the *unique* data value notion satisfying four axioms: Null player, Symmetry, Linearity, and Efficiency. The mathematical definitions of these axioms are deferred to Appendix A.1. Here, we introduce the *linearity* axiom which will be used later.

**Theorem 2** (Linearity of the Shapley value (Shapley, 1953)). *For any of two utility functions $U_1, U_2$ and any $\alpha_1, \alpha_2 \in \mathbb{R}$, we have $\phi_z(\alpha_1 U_1 + \alpha_2 U_2) = \alpha_1 \phi_z(U_1) + \alpha_2 \phi_z(U_2)$.*

**Retraining-based Data Shapley.** The convention of defining the utility function for Data Shapley as $U(S) = \texttt{Perf}(\mathcal{A}(S))$ was introduced in Ghorbani & Zou (2019), where $\mathcal{A}$ is a learning algorithm such as a neural network trained by stochastic gradient descent (SGD) or its variants. With this choice of utility function, the precise calculation of the Shapley value requires retraining models on various subsets of the data. This is because the marginal contribution of a data point, $U(S \cup \{z\}) - U(S)$, can only be obtained by training models on both $S$ and $S \cup \{z\}$ and comparing their performance. As a result, we refer to this method as "*Retraining-based Data Shapley*".

**Limitations beyond efficiency (detailed in Appendix B.1)**: In addition to the high computational costs, we emphasize that retraining-based Data Shapley also suffers from the following limitations: **(1) Highly unstable value scores:** When stochastic learning algorithms such as SGD are used, the resulting value scores can be highly unstable (Wang & Jia, 2023a). This instability may lead to unreliable results and potential violations of Shapley value's fairness axioms. **(2) Conceptual limitations:** Retraining-based Data Shapley measures the average data contribution to the learning process itself, across many retrainings on different data subsets. As a result, it produces attribution scores that apply broadly to the algorithm but fail to reflect the data contribution to a specific training run. On the other hand, providing insights into how individual data points contribute to the deployed model enables more targeted analysis and debugging, thereby improving model interpretability.

## 3 IN-RUN DATA SHAPLEY

To address the issues associated with Retraining-based Data Shapley such as high computational costs, value instability, and the inability to assess the contribution towards a specific trained model, we propose a novel data attribution method specifically tailored for a single training run. Our key idea is to leverage the iterative nature of model training and employ a "divide and conquer" approach: breaking down the problem of valuing data contributions for the entire training process into subproblems of valuing data contributions for individual iterations.

**Utility function for a single gradient update.** Traditionally, the utility function $U(S) = \texttt{Perf}(\mathcal{A}(S))$ encapsulates the overall impact of a training set $S$ across the complete training process.

Here, we instead consider a "local utility function" that evaluates the impact of data subsets within a single iteration. Specifically, given a training dataset $\mathcal{D}_{\mathrm{tr}} = \{z_i\}_{i=1}^N$, a deep learning model is usually being trained to minimize the training loss $\sum_{i=1}^N \ell(w, z_i)$ via an iterative optimization procedure such as SGD. The performance of the model is typically being measured through a set of validation points $\{z^{(\mathrm{val})}\}$. During an iteration $t$, a batch $\mathcal{B}_t \subseteq \mathcal{D}_{\mathrm{tr}}$ of the training points is used to update the model parameters from $w_t$ to $w_{t+1}$ with $w_{t+1} := w_t - \eta_t \sum_{z \in \mathcal{B}_t} \nabla \ell(w_t, z)$, where $\eta_t$ is the learning rate at iteration $t$.[1] A complete run of neural network training thus consists of model checkpoints $\{w_0, w_1, \ldots, w_T\}$. For a given validation data point $z^{(\mathrm{val})}$, we can define the "local utility function" at a single iteration $t$ as

$$U^{(t)}(S; z^{(\mathrm{val})}) := \ell(\widetilde{w}_{t+1}(S), z^{(\mathrm{val})}) - \ell(w_t, z^{(\mathrm{val})}) \tag{1}$$

where $\widetilde{w}_{t+1}(S) := w_t - \eta_t \sum_{z \in S} \nabla \ell(w_t, z)$ and $S \subseteq \mathcal{B}_t$ is a subset of the batch being selected in $t$-th iteration in the original training. **Interpretation:** The local utility function $U^{(t)}$ represents the loss change at iteration $t$ when only the subset $S$ is used for the gradient update. This approach incorporates the realization of random batch selection at $t$-th iteration into the utility function. It can also encode other forms of training randomness (e.g., dropout) at iteration $t$. By accounting for the specific realization of training randomness, we obtain a deterministic utility function for each iteration, effectively enabling the targeted attribution to the specific training run.

**Data Shapley for a single gradient update.** While the utility $U^{(t)}$ is defined over $\mathcal{B}_t$ instead of the full training set $\mathcal{D}_{\mathrm{tr}}$, it is easy to augment it to $\mathcal{D}_{\mathrm{tr}}$. More formally, in the augmented utility function we have $\widetilde{w}_{t+1}(S) := w_t - \eta_t \sum_{z \in S \cap \mathcal{B}_t} \nabla \ell(w_t, z)$, $S \subseteq \mathcal{D}_{\mathrm{tr}}$. The Shapley value $\phi_z(U^{(t)})$ will be exactly the same as the Shapley value corresponds to the augmented utility function for any $z \in \mathcal{B}_t$, and $\phi_z(U^{(t)}) = 0$ for any $z \in \mathcal{D}_{\mathrm{tr}} \setminus \mathcal{B}_t$ (see Theorem 5 in Wang & Jia (2023b)). Therefore, for a clean presentation, we slightly abuse the notation where $U^{(t)}$'s meaning depends on the context.

**Data Shapley for the entire training run.** Building on the concept of a "local" utility function for a single gradient update iteration, we naturally extend this to a "global" utility function for the entire training process, defined as $U(S) = \sum_{t=0}^{T-1} U^{(t)}(S)$. **Interpretation:** This global utility function can be interpreted as the cumulative loss change of the entire training run, but under the counterfactual scenario where only a subset of the training data $S$ is used. In other words, it aggregates the total impact of the subset $S$ on the model's performance throughout the entire training process. Due to the linearity property of the Shapley value (Theorem 2), we have $\phi_z(U) = \sum_{t=0}^{T-1} \phi_z(U^{(t)})$. This new Data Shapley value, which we call *In-Run Data Shapley*, represents the cumulative contribution of the data point $z$ across all gradient update iterations within a single training run. This approach breaks down the broader utility into more manageable, step-by-step assessments that capture the immediate effects of data points on model updates, and provide a more fine-grained view of how individual data points contribute to the model's performance at each step of the training process. Notably, the sum of individual data points' Shapley values equals the overall loss reduction achieved by the model during the entire training run due to the Shapley value's efficiency axiom (see Appendix A.1). This provides a meaningful and interpretable measure of data importance. In Appendix B, we give an in-depth comparison between Retraining-based and In-Run Data Shapley.

**Remark 1 (Multiple validation points).** *In practice, the model performance is often being assessed based on a validation set $D^{(\mathrm{val})} = \{z^{(\mathrm{val})}\}$. After computing $\phi_z\left(U(\cdot; z^{(\mathrm{val})})\right)$ for each $z^{(\mathrm{val})} \in D^{(\mathrm{val})}$, one can compute the Shapley value corresponding to the utility function on the full validation set $U(S; D^{(\mathrm{val})}) := \sum_{z^{(\mathrm{val})} \in D^{(\mathrm{val})}} U(S; z^{(\mathrm{val})})$ by simply taking the sum $\phi_z\left(U(\cdot; D^{(\mathrm{val})})\right) = \sum_{z^{(\mathrm{val})} \in D^{(\mathrm{val})}} \phi_z\left(U(\cdot; z^{(\mathrm{val})})\right)$ due to the linearity property of the Shapley value (Theorem 2). Hence, for a clean presentation, we consider only a single $z^{(\mathrm{val})}$ in this paper. However, all the techniques we developed can be extended to multiple validation points.*

## 4 EFFICIENT COMPUTATION OF IN-RUN DATA SHAPLEY

The newly proposed In-Run Data Shapley does not require retraining models from scratch on different data subsets. However, calculating $\phi_z(U^{(t)})$ for each training iteration remains computationally

---

[1]Note that in practice, we take the gradient average for the update, but here we incorporate the normalization term $|\mathcal{B}_t|$ into the learning rate $\eta_t$ for a clean presentation.

intensive, as it involves evaluating the performance impact of all possible combinations within the sampled data batch. In this section, we introduce an efficient method for approximating In-Run Data Shapley scores during a specific training run. Our approach, distinct from Monte Carlo methods, is deterministic and optimized to minimize additional runtime to regular training. In particular, in its most efficient implementation, our approximation technique incurs negligible extra runtime beyond what is required for standard model training, making it highly practical for real-world applications.

## 4.1 APPROXIMATING $U^{(t)}$ WITH TAYLOR EXPANSION

To derive a more tractable structure for the local utility function $U^{(t)}$, we propose using first and second-order Taylor approximations. The advantage of this approach is that the approximated utility function exhibits a form where closed-form Data Shapley formulas can be derived. The second-order Taylor approximation to the local utility function is as follows:

$$U^{(t)}(S) = \ell(\widetilde{w}_{t+1}(S), z^{(\text{val})}) - \ell(w_t, z^{(\text{val})})$$
$$= \underbrace{\nabla\ell(w_t, z^{(\text{val})}) \cdot (\widetilde{w}_{t+1}(S) - w_t)}_{U^{(t)}_{(1)}(S)} + \frac{1}{2} \underbrace{(\widetilde{w}_{t+1}(S) - w_t)^\mathsf{T} \mathbf{H}^{(z^{(\text{val})})}_t (\widetilde{w}_{t+1}(S) - w_t)}_{U^{(t)}_{(2)}(S)} + \text{higher order terms}$$

where the Hessian matrix $\mathbf{H}^{(z^{(\text{val})})}_t := \nabla^2\ell(w_t, z^{(\text{val})})$. We label the first-order term as $U^{(t)}_{(1)}(S)$ and the second-order term as $U^{(t)}_{(2)}(S)$. Note that the gradient update $\widetilde{w}_{t+1}(S) - w_t = -\eta_t \sum_{z \in S} \nabla\ell(w_t, z)$. Given that the learning rate $\eta_t$ in model training is typically small, a lower-order Taylor expansion often provides an accurate approximation for the change in loss during a single gradient update, with approximation errors of $O(\eta_t^2)$ and $O(\eta_t^3)$ for first and second-order approximations, respectively. In Appendix E.2.2, we empirically investigate the errors of first- and second-order approximations to $U^{(t)}$ on GPT2. In particular, the first-order approximation can already achieve a great performance with Spearman correlation $> 0.94$.

**First-order In-Run Data Shapley.** Using the first-order approximation $U^{(t)} \approx U^{(t)}_{(1)}$, and substituting the gradient update expression, we have $U^{(t)}_{(1)}(S) = -\eta_t \sum_{z \in S} \nabla\ell(w_t, z^{(\text{val})}) \cdot \nabla\ell(w_t, z)$. This shows that $U^{(t)}_{(1)}$ is an *additive* utility function with a closed-form Shapley calculation as follows:

**Theorem 3.** *In-Run Data Shapley considering the first-order approximation has closed-form*

$$\phi_z(U) \approx \sum_{t=0}^{T-1} \phi_z\left(U^{(t)}_{(1)}\right)$$

*where*

$$\phi_z\left(U^{(t)}_{(1)}\right) = -\eta_t \nabla\ell(w_t, z^{(\text{val})}) \cdot \nabla\ell(w_t, z), \quad t = 0, \ldots, T-1$$

Thus, the first-order approximation of In-Run Data Shapley for a training point accumulates its gradient dot products with the validation data point each time the training point is sampled in the training batch. The gradient dot product between the training point $z_i$ and the validation point $z^{(\text{val})}$ represents the direct influence of $z_i$ on the validation loss at the current model parameters $w_t$, which essentially measures the alignment between the two gradient vectors in the parameter space. Notably, $\phi_z\left(U^{(t)}_{(1)}\right)$ is equivalent to the TracIN-Ideal score proposed by (Pruthi et al., 2020). That is, the TracIN-Ideal score can be interpreted as the Shapley value when we use first-order Taylor approximation for $U^{(t)}$. However, TracIN-Ideal has been described as "computationally infeasible," and our approach completely overcomes this problem. In Appendix A.3, we provide a detailed discussion of the differences between this work and (Pruthi et al., 2020).

**Second-order In-Run Data Shapley.** We further improve the approximation of $U^{(t)}$ using a second-order Taylor expansion, i.e., $U^{(t)} \approx U^{(t)}_{(1)} + \frac{1}{2}U^{(t)}_{(2)}$. Fortunately, the approximated utility function maintains a tractable structure that allows a closed-form Shapley value calculation.

**Theorem 4.** *In-Run Data Shapley considering the second-order approximation has closed-form*

$$\phi_z(U) \approx \sum_{t=0}^{T-1}\left(\phi_z\left(U^{(t)}_{(1)}\right) + \frac{1}{2}\phi_z\left(U^{(t)}_{(2)}\right)\right) \tag{2}$$

*where*

$$\phi_z\left(U_{(1)}^{(t)}\right) + \frac{1}{2}\phi_z\left(U_{(2)}^{(t)}\right) = \underbrace{-\eta_t \nabla \ell(w_t, z^{(\mathrm{val})}) \cdot \nabla \ell(w_t, z)}_{\textcircled{1} \text{ influence of } z \text{ on the loss of } z^{(\mathrm{val})}} + \underbrace{\frac{\eta_t^2}{2}\nabla \ell(w_t, z)^\intercal H_t^{(z^{(\mathrm{val})})}\left(\sum_{z_j \in \mathcal{B}_t} \nabla \ell(w_t, z_j)\right)}_{\textcircled{2} \text{ interaction between } z \text{ and other training points}} \quad (3)$$

*for any $t = 0, \ldots, T-1$.*

Compared to the first-order variant, the second-order In-Run Data Shapley includes an additional gradient-Hessian-gradient product term that captures the interaction between the training point of interest $z$ and the rest of the training set. The Hessian matrix represents the curvature of the validation loss function at the current model parameters $w_t$. This interaction term measures the alignment between the gradient of $z$ and the gradients of the other points in the training batch, adjusted by the Hessian. If this term is large, it indicates that the presence of other points in the batch significantly impacts the value attributed to $z$. For example, if there are many identical or similar copies of $z$ in the training set, the contribution of $z$ will decrease, as the interaction term will be large, effectively distributing the value among the similar points. By incorporating this interaction term, the second-order In-Run Data Shapley provides a more fine-grained contribution measure that takes into account both the relevance of a data point towards a validation set and its uniqueness within the population.

### 4.2 Efficient Computation of Gradient Dot-product and Gradient-Hessian-Gradient Product

Although we have derived closed-form formulas for In-Run Data Shapley using first- or second-order Taylor approximation of the local utility functions, efficiently computing these values remains a challenge. Specifically, for the first-order In-Run Data Shapley, it requires computing $\textcircled{1}$ the pairwise gradient dot products between each $z \in \mathcal{B}_t$ and the validation point. For the second-order In-Run Data Shapley, it additionally requires computing $\textcircled{2}$ the gradient-Hessian-gradient products for each $z \in \mathcal{B}_t$. A direct implementation to compute $\textcircled{1}$ involves calculating the individual gradient for each data point in $\mathcal{B}_t$, which cannot benefit from fast batch processing in GPUs and necessitates running backpropagation $|\mathcal{B}_t|$ times with a mini-batch size of 1. Consequently, this approach would be at least $|\mathcal{B}_t|$ times slower than regular training, making it computationally prohibitive for practical applications. Furthermore, computing $\textcircled{2}$ requires either computing each individual gradient again or storing all individual gradients, which incurs significant time or memory overhead.

**Computing pair-wise gradient dot-products in 1 backpropagation.** Our technique for efficiently computing pairwise gradient dot products is inspired by the "ghost clipping" technique from the differential privacy (DP) literature (Lee & Kifer, 2021). "Ghost clipping" enables computing *all* of the per-sample gradient norms within one backpropagation without explicitly forming any individual gradient vectors, which enhances the efficiency of DP model training. Here, we propose a "ghost dot-product" technique that shares the idea of exploiting the computation that has been done in the backpropagation. Specifically, denote a sample batch as $\mathcal{B}_t = \{z_1, \ldots, z_B\}$. We demonstrate this technique using a simple linear layer $\mathbf{s} = \mathbf{a}\mathbf{W}$, where $\mathbf{W} \in \mathbb{R}^{d_1 \times d_2}$ is the weight matrix, $\mathbf{a} = (\mathbf{a}^{(1)}, \ldots, \mathbf{a}^{(B)})^\intercal$ is the mini-batch input, and $\mathbf{s} = (\mathbf{s}^{(1)}, \ldots, \mathbf{s}^{(B)})^\intercal$ is the output (i.e., the pre-activation tensor). For (non-sequential) data, $\mathbf{a} \in \mathbb{R}^{B \times d_1}, \mathbf{s} \in \mathbb{R}^{B \times d_2}$. By applying the chain rule, we can express the gradient of an individual loss $\ell^{(i)} := \ell(w, z_i)$ with respect to $\mathbf{W}$ as

$$\frac{\partial \ell^{(i)}}{\partial \mathbf{W}} = \frac{\partial \ell^{(i)}}{\partial \mathbf{s}^{(i)}} \otimes \frac{\partial \mathbf{s}^{(i)}}{\partial \mathbf{W}} = \frac{\partial \ell^{(i)}}{\partial \mathbf{s}^{(i)}} \otimes \mathbf{a}^{(i)} = \frac{\partial \ell}{\partial \mathbf{s}^{(i)}} \otimes \mathbf{a}^{(i)} \quad (4)$$

where $\ell := \sum_{j=1}^B \ell^{(j)}$ is the aggregated loss, and the last step is because other data points' losses have no dependency on $\mathbf{s}_i$. Note that the individual's output gradient $\frac{\partial \ell^{(i)}}{\partial \mathbf{s}^{(i)}} = \frac{\partial \ell}{\partial \mathbf{s}^{(i)}}$ is readily available during the backpropagation pass in terms of $\ell$. Suppose we are interested in computing the gradient dot-product $\frac{\partial \ell^{(1)}}{\partial \mathbf{W}} \odot \frac{\partial \ell^{(2)}}{\partial \mathbf{W}}$ between two data points $z_1, z_2$ in the same batch in the backpropagation. For non-sequential data, we have each $\mathbf{a}^{(i)} \in \mathbb{R}^{d_1 \times 1}$ and $\frac{\partial \ell^{(i)}}{\partial \mathbf{s}^{(i)}} \in \mathbb{R}^{1 \times d_2}$. By (4), we have

$$\frac{\partial \ell^{(1)}}{\partial \mathbf{W}} \odot \frac{\partial \ell^{(2)}}{\partial \mathbf{W}} = \left(\mathbf{a}^{(1)} \otimes \frac{\partial \ell^{(1)}}{\partial \mathbf{s}^{(1)}}\right) \odot \left(\mathbf{a}^{(2)} \otimes \frac{\partial \ell^{(2)}}{\partial \mathbf{s}^{(2)}}\right) = \left(\left(\mathbf{a}^{(1)}\right)^\intercal \mathbf{a}^{(2)}\right)\left(\left(\frac{\partial \ell^{(1)}}{\partial \mathbf{s}^{(1)}}\right)^\intercal \left(\frac{\partial \ell^{(2)}}{\partial \mathbf{s}^{(2)}}\right)\right) \quad (5)$$

Hence, we can first take the two inner products, and then multiply the results together. All of the quantities $\mathbf{a}^{(1)}, \mathbf{a}^{(2)}, \frac{\partial \ell^{(1)}}{\partial \mathbf{s}^{(1)}}, \frac{\partial \ell^{(2)}}{\partial \mathbf{s}^{(2)}}$ in (5) that are required for computation are all already available in the backpropagation. Hence, within a *single* backpropagation, we can efficiently compute the gradient dot-product between *every* pair of $z_i, z_j \in \mathcal{B}_t$. Since we are interested in computing the gradient dot-product between $z^{(\text{val})}$ and $z$ for all $z \in \mathcal{B}_t$, we can backpropagate on $\sum_{z \in \mathcal{B}_t} \ell^{(i)} + \ell^{(z^{(\text{val})})}$ to save another backpropagation for $z^{(\text{val})}$. We call this technique the *"ghost dot-product"*, as no gradient vectors are instantiated during the computation. Overall, we only need *one* backpropagation to compute ① for *all* data points in $\mathcal{B}_t$, a significant improvement over the direct method requiring $\geq |\mathcal{B}_t|$ backpropagations. Additional details for this technique are in Appendix D.

**Remark 2.** *While we illustrate our "ghost dot-product" technique using linear layers, it can be extended to other types of layers by leveraging similar decompositions as in Equation (4) that have been developed in differential privacy literature (Rochette et al., 2020; Bu et al., 2022; Li et al., 2021; Bu et al., 2023; Kong & Munoz Medina, 2024).*

**Computing gradient-Hessian-gradient products in 2 backpropagations (Appendix D.2).** For second-order In-Run Data Shapley, an outstanding challenge is how to efficiently compute the pairwise interaction term among training points. In Appendix D.2, we develop a *"ghost gradient-Hessian-gradient product"* technique for computing the desired quantity through one extra backpropagation pass, without materializing any gradient-sized vectors. This technique leverages several properties of neural network gradients across different layers, and its derivation is complex.

**Further improvement of runtime and memory requirements (Appendix D.3).** With the "ghost" techniques developed, the computation of first- and second-order In-Run Data Shapley requires one and two backpropagations in each gradient update iteration respectively. Although we still need to compute the gradient of the aggregated loss $\sum_{z_i \in \mathcal{B}_t} \ell_i$ for the training batch to perform parameter updates, we do *not* need an additional backpropagation. By reusing the activations and output gradients from the previous backpropagation on $\sum_{z \in \mathcal{B}_t} \ell^{(i)} + \ell^{(z^{(\text{val})})}$, we can easily compute this quantity *without* incurring the cost of an extra backpropagation pass. Consequently, training while computing first-order In-Run Data Shapley will have minimal additional runtime overhead, as it still requires only one backpropagation per iteration. The second-order In-Run Data Shapley necessitates one extra backpropagation per iteration. Nevertheless, both methods provide significant advantages over the direct approach of instantiating per-sample gradients and Hessian-vector products.

## 5 EXPERIMENTS

In this section, we evaluate In-Run Data Shapley in terms of its efficiency (Section 5.1), fidelity (Section 5.2), and its applications in data attribution for language model pretraining (Section 5.3). In Appendix E.6, we compare a variety of existing data attribution methods in small-scale settings.

### 5.1 RUNTIME EVALUATION

We empirically assess the computational efficiency of In-Run Data Shapley with "ghost dot-product" and "ghost vector-Hessian-vector product" techniques developed in Section 4.2. We compare this to the direct implementation of In-Run Data Shapley, which requires computing per-sample gradients, as well as to regular training without Data Shapley computations. The experiment is conducted by training GPT2-Small on a single 80GB A100 GPU. As illustrated in Table 1, the runtime of first-order In-Run Data Shapley is close to that of regular training when using the ghost dot-product algorithms developed in Section 4.2. The second-order In-Run Data Shapley is approximately $2\times$ slower than regular training due to the additional backpropagation. However, both the first- and second-order In-Run Data Shapley are significantly faster ($> 30\times$) compared to the naive implementation. These results showcase the substantial improvements achieved by our techniques, making In-Run Data Shapley computationally feasible for practical applications.

|  | Throughput |
|---|---|
| **Regular Training** | 76.2 |
| **First-order Data Shapley (ghost)** | 70.5 |
| **Second-order Data Shapley (ghost)** | 34.4 |
| **First-order Data Shapley (direct)** | 4.2 |
| **Second-order Data Shapley (direct)** | 1.8 |

Table 1: Efficiency comparison of different implementations of In-Run Data Shapley. We use throughput, i.e., # training data points being processed per second as the efficiency metric.

## 5.2 FIDELITY EVALUATION

In this section, we directly assess the approximation accuracy of first- and second-order In-Run Data Shapley. In Appendix E.6, we further compare the performance of In-Run Data Shapley with existing (less scalable) data attribution techniques on standard benchmarks (e.g., mislabeled data detection) as an additional sanity check.

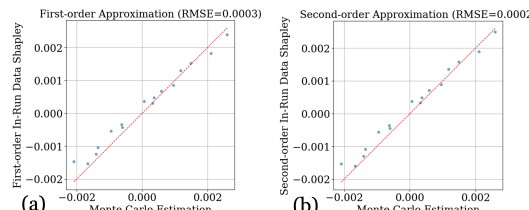

Figure 1: Comparison between the Monte Carlo-estimated In-Run Data Shapley and First/Second-order In-Run Data Shapley.

Given that computing the exact In-Run Data Shapley value is computationally prohibitive, we compare their performance against Monte Carlo estimates of the Shapley value using a large number of samples. The experiment is conducted on GPT2 model at the 3500th training iteration on Pile, where the batch size 16 and learning rate $3 \times 10^{-4}$. We use 1000 permutations to approximate the groundtruth In-Run Shapley value. Figure 1 shows that with just first-order In-Run Data Shapley, the root mean squared error (RMSE) is only around 0.0003. In Appendix E.2.1, we provide additional results with different learning rates.

## 5.3 CASE STUDY: DATA ATTRIBUTION ON PILE DATASET

In this section, we present a case study to demonstrate the use cases of In-Run Data Shapley by pretraining on the well-known Pile dataset (Gao et al., 2020). We explore its application in data curation, examine data contribution across different training stages, and investigate relevant corpus detection. Due to computational resource constraints, most of our experiments focus on the GPT-2 and Pythia-410M models, but this is not a limitation of the algorithm itself. With adequate computational resources, our approach can easily be applied to larger-scale models.

### 5.3.1 IS WELL-CURATED DATASET ACTUALLY CLEAN?

Carefully curated pretraining corpora still contain data points that can adversely affect the training process. Identifying and removing these data points can accelerate model convergence and enhance overall performance, thereby saving computational resources. In this experiment, we demonstrate the effectiveness of In-Run Data Shapley in assessing the data quality of a subset of the Pile dataset. We uniformly select a random subset of Pile with around 10B tokens and train a GPT2 model on this subset. We compute the data attribution results with Pile's validation set. By filtering out all negatively valued corpora and retraining the model on the cleaned subset, we observe significant improvement in model convergence. For both first- and second-order In-Run Data Shapely, we can achieve around 25% fewer training iterations to reach a test loss of 3.75. Surprisingly, our analysis reveals that around 16% of the training corpora had negative second-order In-Run Data Shapley values. While some of the negatively valued corpora may be attributed to the significant domain shift compared to the validation corpora, we still find many low-quality corpora from Pile, a pretraining dataset that has undergone several layers of data curation (Gao et al., 2020). Examples of low-quality corpora identified can be found in Appendix E.3.

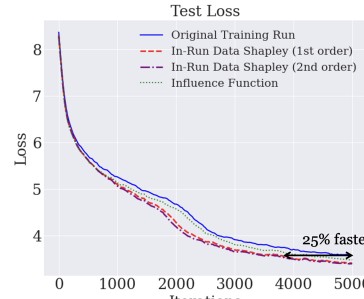

Figure 2 shows a performance comparison between the original training run and the model trained on the cleaned subsets for GPT2, and additional results on Pythia-410M are available in Appendix E.3.1. We also compare with influence function (Koh & Liang, 2017), which approximates the change in the model's loss on the test example when the training example is removed from the training set (i.e., the leave-one-out score). We omit TRAK (Park et al., 2023) and other techniques such as datamodel (Ilyas et al., 2022) as they are not scalable to our setting. As we can see, influence function can also filter out low-quality data that can accelerate training convergence. However, the performance is slightly worse than In-Run Data Shapley as the influence function only uses information from the final trained models, which can result in highly noisy value scores since the removal of one training data point might have a negligible effect on the final model performance.

Figure 2: Test loss comparison between the original training run and the model trained on the cleaned subset according to different data attribution techniques.

The results demonstrate that removing lower-quality

| **Original Wikipedia Corpus** | **Synthetic "Similar topic" Corpus** |
|---|---|
| In 2012, Radhi recruited new 'musicians' for OAG, who were selected from among the students of Akademi Seni Budaya dan Warisan Kebangsaan (). The new line-up consists of Qi Razali (drums/backing vocals - original drummer ... | ### Instruction: Write a short story about a classical violinist who decides to explore jazz music, detailing her first encounter with a jazz band. ### Answer: Elena, a classically trained violinist known for her precise and emotive performances ... |

| Similarity Category | In-Run Data Shapley (1st order) | In-Run Data Shapley (2nd order) | Influence Function | BM25 |
|---|---|---|---|---|
| **Partial exactly the same** | 1 | 1 | 1 | 1 |
| **Paraphrase** | 1 | 1 | 1 | 1 |
| **Significant paraphrase** | 32.3 | 32 | 39.3 | 1.6 |
| **Similar topic** | 145.6 | 141.6 | 292 | 20917.3 |

Table 2: Top: (left) An original training corpus from Wikipedia. (right) A synthetic corpus falls in the category of "Similar topic" to the Wikipedia corpus on the left (prompt in Appendix E.5). Bottom: the (average) value rank of the original corpus among all training corpora for validation corpora that are of varying similarity to the original corpus. **The rank is out of ≈320k data points.**

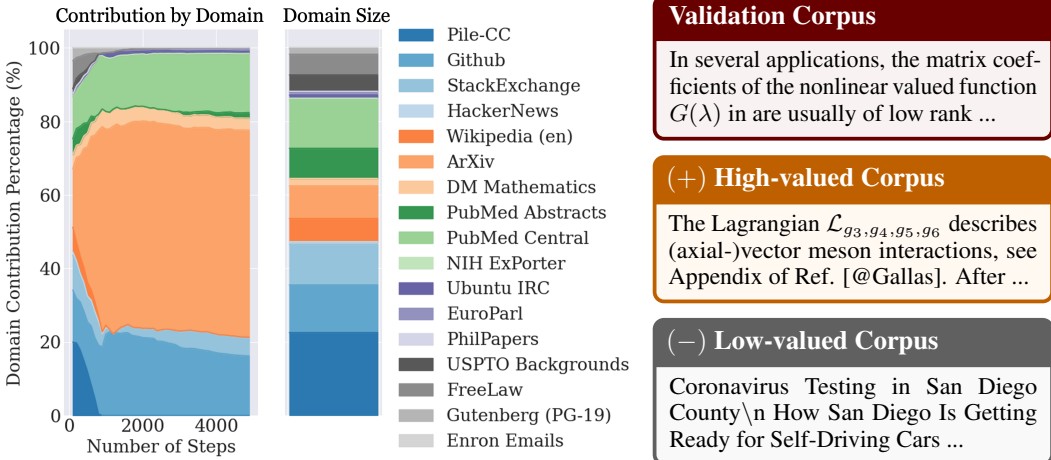

Figure 3: Left: Domain value composition for a corpus of math text. Right: The math corpus we use as the validation data for attribution, and examples of high- and low-valued training corpus for it.

data leads to a significantly faster drop in test loss compared to the original training run. This implies that there is still huge room for data curation for well-curated datasets such as Pile.

### 5.3.2 How Do Data Values Change during Training?

As In-Run Data Shapley tracks the cumulative data values across different training steps, we can assess the contribution of training points at various stages of training, providing a more fine-grained perspective on data attribution. We evaluate the data attribution results for a math-related validation data using second-order In-Run Data Shapley. In Figure 3, we present the value composition of training corpora by their domains over the first 10,000 training iterations, summing the values of all corpora from the same domain. We then calculate the percentage of the total value attributed to each domain, excluding domains with a total value < 0. As illustrated in the figure, the corpora from ArXiv achieve a significantly higher value compared to other domain corpora, far exceeding its size proportion within the full Pile dataset. This is expected, as ArXiv papers predominantly cover fields like Math, Computer Science, and Physics, which contain extensive math-related content. Furthermore, the value composition changes rapidly at the beginning of training and stabilizes as training progresses. We hypothesize that this initial fluctuation is due to the presence of relevant paragraphs in corpora from other domains. The stable value proportion observed in later stages likely reflects the relative abundance of math content in each domain. Interestingly, we observe that Pile-CC domain, which contains general website crawls, initially shows positive contributions during the first few iterations. However, its value quickly drops to negative and eventually converges to zero. This implies that **general corpora tend to have a large contribution in the beginning of training**, as they help the model learn the basics of languages, such as grammar and common knowledge. However, as training progresses and the model focuses on more specialized topics, the relevance of general domains diminishes. An additional figure for the average domain values is in Appendix E.4.

### 5.3.3 Does Contribution Require Memorization?

In this experiment, we evaluate the robustness of different data attribution techniques in identifying relevant individual corpora that have been paraphrased. We start by selecting a data point from the training set and creating several paraphrased versions using GPT-4, with varying levels of paraphrasing (see Appendix E.5 for the prompt). These paraphrased versions form our validation set. We then calculate the average value rank of the original training data for each of its paraphrased versions. In addition to In-Run Data Shapley and influence function, we include the ranking result based on BM25 distance. **BM25** (Robertson et al., 2009) featurizes examples by their word frequency statistics (i.e., TF-IDF) to rank the training instances. We use BM25 distance as an oracle for assessing the verbatim or lexical similarity between the validation data (query) and the training data, as opposed to semantic similarity.

As shown in Table 2, even for a validation data that is a complete rewrite (with a low BM25 distance) but covers relevant topics, the original training data still ranks very high according to both In-Run Data Shapley and influence function. Influence function ranks the original training data lower than In-Run Data Shapley, which may be attributed to the inherent noisy nature of the leave-one-out error estimation. The results of this experiment have important implications for the ongoing discussion about the copyright of generative AI. Specifically, the table presents a compelling example where the original Wikipedia training corpus related to a musician's experience can significantly contribute to generating a story about a musician, even when the generated story shares no token-wise resemblance to the original training data. This finding supports that training data profoundly influences the capabilities of generative AI models and should be compensated accordingly (Deng & Ma, 2023; Wang et al., 2024b), even when the output does not closely resemble the original copyrighted material or when the model applies output filters to avoid generating verbatim replicates of the training data. This discovery expands the conventional focus on copyright violations, which typically addresses instances of near-verbatim replication, as seen in the dispute between New York Times and OpenAI, to also include cases where the generated content is significantly influenced by copyrighted material without directly replicating it.

## 6 Conclusion and Limitations

In this work, we introduce In-Run Data Shapley, a novel data attribution technique that addresses the limitations of Retraining-based Data Shapley. Extensive experiments demonstrate the effectiveness of In-Run Data Shapley in various applications. Here, we discuss the potential limitations of this work.

**Availability of validation data before training.** One potential limitation of In-Run Data Shapley is that it requires the validation data to be available before training, as the data attribution scores are computed during the training process. However, there are many scenarios where validation data is naturally available before training, such as when using publicly available benchmark datasets, participating in machine learning competitions, or adhering to regulatory requirements. For scenarios where validation data arrives after the model has been trained, a potential solution is to save checkpoints of intermediate models during training and approximate In-Run Data Shapley using these checkpoints, in the same spirit of TracIN-CP described in Pruthi et al. (2020). However, the choice of checkpoints can significantly impact performance, and it is unclear which checkpoints to pick.

**Extension to other optimization algorithms.** Extending the "ghost" family techniques developed in this work to support Adam and similar optimizers remains an exciting direction for future research. We stress that extending the formulation of In-Run Data Shapley from SGD to Adam is feasible, but the actual challenge lies in computing it efficiently without instantiating each individual gradient vector, which cannot be solved by simple extensions described in Xia et al. (2024).

**Handling memory constraints.** In scenarios where GPU memory constraints prevent large batch sizes, the "ghost" techniques can be extended by using gradient accumulation. This approach accommodates larger training batch sizes by dividing the batch into smaller sub-batches and accumulating gradients over multiple iterations. While this method may increase runtime due to additional back-propagation steps, it maintains the feasibility of the techniques under memory constraints. Improving computational efficiency for large batch sizes remains an important direction for future research.

ACKNOWLEDGMENT

This work is supported in part by the National Science Foundation under grants IIS-2312794, IIS-2313130, OAC-2239622, Amazon-Virginia Tech Initiative in Efficient and Robust Machine Learning, and the Commonwealth Cyber Initiative.

We thank Meng Ding, Chong Xiang, Chendi Wang for their helpful feedback on the preliminary version of this work.

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

## A  EXTENDED RELATED WORKS

### A.1  DATA SHAPLEY AXIOMS

*Data Shapley* is one of the first principled approaches to data attribution being proposed Ghorbani & Zou (2019); Jia et al. (2019b). Data Shapley is based on the famous *Shapley value* (Shapley, 1953). In almost all of the literature, the Shapley value is being justified as the *unique* value notion satisfying the following four axioms:

1. **Null player:** if $U(S \cup \{z_i\}) = U(S)$ for all $S \subseteq D \setminus \{z_i\}$, then $\phi_{z_i}(U) = 0$.
2. **Symmetry:** if $U(S \cup \{z_i\}) = U(S \cup \{z_j\})$ for all $S \subseteq D \setminus \{z_i, z_j\}$, then $\phi_{z_i}(U) = \phi_{z_j}(U)$.
3. **Linearity:** For utility functions $U_1, U_2$ and any $\alpha_1, \alpha_2 \in \mathbb{R}$, $\phi_{z_i}(\alpha_1 U_1 + \alpha_2 U_2) = \alpha_1 \phi_{z_i}(U_1) + \alpha_2 \phi_{z_i}(U_2)$.
4. **Efficiency:** for every $U, \sum_{z_i \in D} \phi_{z_i}(U) = U(D)$.

In plain words, **null player** axiom means the Shapley value will assign zero score to data points with no contribution. **Symmetry** axiom requires equal scores assigned to equally impactful data points, ensuring fairness. **Efficiency** axiom requires the sum of contribution scores equal to the total utility, meaning the scores always represent a share of the total utility. **Linearity** axiom means the Shapley value supports additive decomposition across multiple utility functions, allowing for the calculation of contributions to the entire test set by summing the scores of individual test points.

### A.2  DATA SHAPLEY AND FRIENDS

Since its introduction in 2019 (Ghorbani & Zou, 2019; Jia et al., 2019b), Data Shapley has rapidly gained popularity as a principled solution for data attribution. Due to the computationally expensive nature of retraining-based Data Shapley, various Monte Carlo-based approximation algorithms have been developed (Jia et al., 2019b; Illés & Kerényi, 2019; Okhrati & Lipani, 2021; Burgess & Chapman, 2021; Mitchell et al., 2022; Lin et al., 2022; Wang & Jia, 2023b; Li & Yu, 2023; Covert et al., 2024), these methods still necessitate extensive computational resources due to repeated model retraining, which is clearly impractical for modern-sized ML models. Many of its variants have been proposed. Kwon & Zou (2022) argues that the efficiency axiom is not necessary for many machine learning applications, and the framework of *semivalue* is derived by relaxing the efficiency axiom. Lin et al. (2022) provide an alternative justification for semivalue based on causal inference and randomized experiments. Based on the framework of semivalue, Kwon & Zou (2022) propose *Beta Shapley*, which is a collection of semivalues that enjoy certain mathematical convenience. Wang & Jia (2023a) propose *Data Banzhaf*, and show that the Banzhaf value, another famous solution concept from cooperative game theory, achieves more stable valuation results under stochastic learning algorithms. Li & Yu (2024) further improves the valuation stability by considering value notions outside the scope of semivalue.

The classic leave-one-out error is also a semivalue, where the *influence function* (Cook & Weisberg, 1980; Koh & Liang, 2017; Grosse et al., 2023) is generally considered as its approximation. A concurrent work (Choe et al., 2024) leverages a similar gradient decomposition technique as our paper to speed up influence function calculation. However, several works have pointed out the fragility of influence function for deep learning models (Basu et al., 2021; Søgaard et al., 2021; Bae et al., 2022), due to the strong assumptions of training convergence and strongly convexity of the loss function. Nguyen et al. (2023) takes a Bayesian view of data attribution and is able to evaluate the variance of LOO. Unlike Nguyen et al. (2023), our work explicitly incorporates specific training randomness (e.g., model initialization, batch ordering) into the utility function definition, providing attribution scores that reflect contributions to the exact training trajectory.

Another line of works focuses on improving the computational efficiency of Data Shapley by considering K nearest neighbor (KNN) as the surrogate learning algorithm for the original, potentially complicated deep learning models (Jia et al., 2019a; Wang & Jia, 2023c; Wang et al., 2023; 2024c; Yang et al., 2024). Ghorbani et al. (2020); Kwon et al. (2021); Li & Yu (2023) consider Distributional Shapley, a generalization of Data Shapley to data distribution. In federated learning setting, Wang et al. (2020) proposes a similar idea of computing the Shapley value for each federated learning round.

**Remark 3.** *Randomized Monte Carlo estimators can be inefficient and may produce unstable valuation results, potentially violating fairness axioms of the Shapley value (e.g., the Symmetry axiom) due to inherent approximation errors. In contrast, In-Run Data Shapley does not rely on Monte Carlo estimators. Instead, it computes the exact Shapley value for an approximated utility function via Taylor expansion. This approach adheres to the fairness axioms while ensuring the reliability and consistency of the data valuation results.*

### A.3 COMPARISON WITH TRACIN (PRUTHI ET AL., 2020)

The form of first-order In-Run Data Shapley from Section 4.1 coincides with the TracIN-Ideal in Pruthi et al. (2020). This provides a new understanding of TracIN-Ideal as an approximation to In-Run Data Shapley. Both works face the technical challenge of requiring per-sample gradient computations during a single training run. Pruthi et al. (2020) proposes *TracIN-CP*, which mitigates the computational burden by examining only a subset of intermediate checkpoints during training. At each checkpoint, the individual gradients for the entire training set are computed, rather than for a sampled batch, under the assumption that each training example is visited exactly once between checkpoints. A recent work (Xia et al., 2024) leverages TracIN-CP, an approximation algorithm for TracIN-Ideal, for instruction-following data selection. This approach, however, may deviate significantly from the original TracIN-Ideal, with the final valuation results heavily dependent on the selected checkpoints. Furthermore, Pruthi et al. (2020)'s implementation is limited to the parameters of the last linear layer due to memory constraints, potentially biasing the measurement of data contribution. For instance, Yeh et al. (2022) suggests that the last layer of a neural network might exhibit a strong "cancellation effect," where the data influence of different examples have large, contradictory magnitudes. Additionally, Schioppa (2024) demonstrates that selecting different layers can distort data attribution scores. In contrast, this work introduces the "ghost dot-product" technique to efficiently compute the first-order In-Run Data Shapley (i.e., TracIN-Ideal) directly and accurately, without additional approximations.

# B ADDITIONAL DISCUSSION

In this section, we provide additional discussion about In-Run Data Shapley as well as its comparison with Retraining-based Data Shapley. Figure 4 gives a more detailed overview of our algorithm, and Figure 5 provides a visualized comparison between Retraining-based and In-Run Data Shapley.

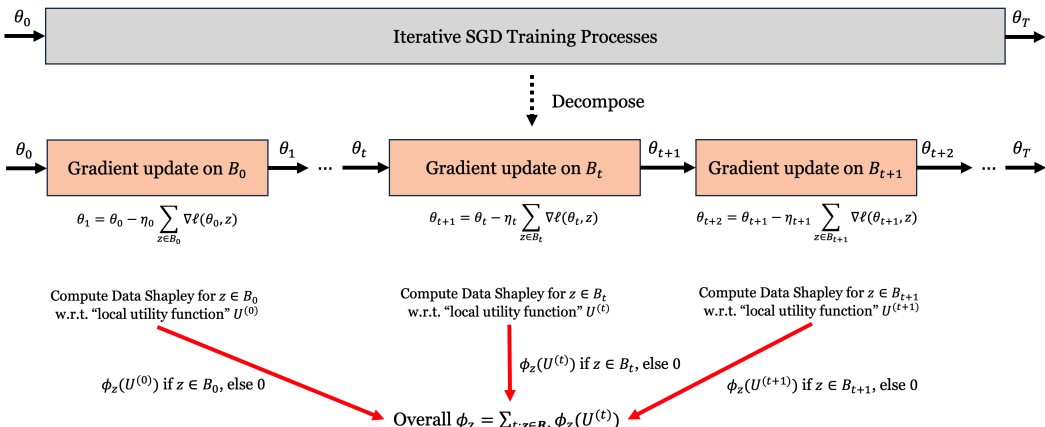

Figure 4: The core idea and algorithm overview of In-Run Data Shapley. Rather than evaluating data contribution across the entire training process (top), we decompose it into individual gradient update steps (bottom). For each iteration $t$, we compute the Shapley value $\phi_z(U^{(t)})$ with respect to a "local utility function" $U^{(t)}$ that measures how the batch $\mathcal{B}_t$ contributes to reducing validation loss. A data point's final contribution score is the sum of its Shapley values across all iterations where it appears: $\phi_z = \sum_{t:z \in \mathcal{B}_t} \phi_z(U^{(t)})$. This decomposition approach maintains the Shapley value properties through the linearity axiom while making computation tractable.

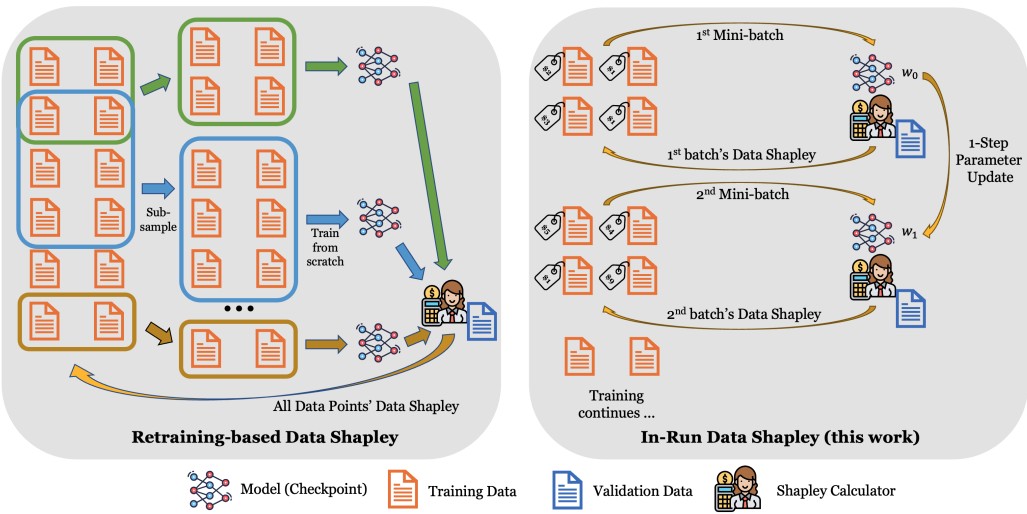

Figure 5: Comparison between Retraining-based and In-Run Data Shapley. Retraining-based Data Shapley requires training a model from scratch on all possible subsets of the full training set, which is computationally inefficient and raises concerns about interpretability and stability. In contrast, In-Run Data Shapley acts as a "contribution accountant", efficiently tracking and attributing data value scores to each training example across gradient update steps during a single training run.

### B.1   WHEN IS IN-RUN DATA SHAPLEY DESIRABLE FOR DATA ATTRIBUTION?

While Retraining-based Data Shapley has been widely adopted in the literature, it suffers from several critical issues that limit its practicality and effectiveness. In this section, we discuss these problems from four key aspects: computational efficiency, alignment with the purpose of data valuation, stability of the valuation results, and the choice of training hyperparameters.

**(1) Computational burden.** Retraining-based Data Shapley calculation is often computationally prohibitive, as it requires retraining the model on every possible subset of the original dataset, leading to a computational complexity that grows exponentially with the size of the dataset. Despite the development of various Monte Carlo-based approximation algorithms (Jia et al., 2019b; Illés & Kerényi, 2019; Okhrati & Lipani, 2021; Burgess & Chapman, 2021; Mitchell et al., 2022; Lin et al., 2022; Wang & Jia, 2023b; Li & Yu, 2023; Covert et al., 2024), these methods still necessitate extensive computational resources due to repeated model retraining, which is clearly impractical for modern-sized ML models. Another line of work attempts to use efficient proxy learning algorithms, such as $K$-nearest neighbors (KNN) (Jia et al., 2019a; Wang & Jia, 2023c; Wang et al., 2023; 2024c; Yang et al., 2024), to accelerate Data Shapley computation. However, it remains unclear how closely these cheaper proxy models approximate the original learning algorithm, and it is also uncertain how to interpret the derived Data Shapley scores in this context.

**(2) Retraining-based Data Shapley is unable to assess data contribution to a specific model.** Crucially, Retraining-based Data Shapley is not designed to value data contribution towards a specific model. It attempts to quantify the average contribution of each training data point to models trained on different subsets of the data, rather than its contribution to the specific model trained on the full dataset. While one might interpret the former as an approximation of the latter, these two quantities can be quite different in practice, especially when the learning algorithm is randomized and sensitive to factors like random initialization and the order of the data points during training. More importantly, in most real-life scenarios, the primary interest lies in understanding the contribution of each data point to the specific model being trained and deployed.

**(3) Retraining-based Data Shapley produces unstable valuation results for stochastic training algorithms.** Furthermore, when the training algorithm involves randomness, such as in the case of SGD with random mini-batch selection, the corresponding utility function becomes randomized. Prior work (Wang & Jia, 2023a) suggests that this randomness can introduce substantial noise into the estimated Shapley values, rendering them unreliable and unstable. This instability poses significant challenges for interpreting and using the resulting data valuations, as the scores may vary considerably across different runs of the algorithm, even on the same dataset. Consequently, this limits the practical applicability of Retraining-based Data Shapley when working with stochastic training algorithms, which are prevalent in modern machine learning. We note that similar vulnerabilities to learning stochasticity have been observed for LOO-based data influence scores (e.g., influence function (Koh & Liang, 2017)) in several works (Basu et al., 2021; Søgaard et al., 2021; Nguyen et al., 2023).

**(4) Training hyperparameter choices in Retraining-based Data Shapley are unclear.** In machine learning, training hyperparameters (e.g., learning rate and batch size) typically need to be adjusted based on the size of the training dataset. This creates a *fundamental ambiguity* when computing Data Shapley values: when evaluating the utility $U(S)$ for different data subsets $S$, should we use the same hyperparameters as those optimized for the full dataset, or should we adjust them based on the subset size? This choice can significantly impact the calculated values.

In-Run Data Shapley addresses all these issues, making it more desirable for specific scenarios. Firstly, it is computationally efficient as it computes data values during the training run, avoiding the need for multiple retraining iterations. This makes it feasible for modern large-scale ML models. Secondly, it aligns better with the purpose of data valuation by assessing the contribution of each data point to the specific model being trained, providing more relevant insights for real-world applications. Lastly, In-Run Data Shapley offers deterministic valuation results even with stochastic training algorithms, as it incorporates the specific sequence and randomness of the training process into the utility function. Therefore, for scenarios where computational efficiency, model-specific contributions, and result stability are critical, In-Run Data Shapley is a more suitable choice.

**Remark 4** (In-run Data Shapley is a model-specific data attribution technique.). *In the original Data Shapley literature, $U(S)$ typically defines utility as the final performance (e.g., accuracy, loss) of a model trained on a data subset $S$. However, things become more complicated in the context of deep learning. A specific deep learning training run is an* iterative process. *Defining utility $U(S)$ for a particular run makes it a* sequence function *rather than the pure* set function *required by the standard Shapley value framework. Extending Shapley axioms to sequence functions is not straightforward. Previous works often define $U(S)$ as "expected model utility across all possible training runs" to circumvent this conceptual issue. In this work, we initiate the study of* model-specific data attribution, *which is crucial for applications like model diagnosis and behavior interpretation where we focus on a specific training run rather than expected utility across all possible runs.*

***Data attribution should consider specific application scenario.*** *From a broader perspective, the "optimal data attribution" notion heavily depends on the intended application. In the literature, data attribution techniques are being used for various purposes such as data quality assessment, data valuation, and interpreting model predictions. For data quality assessment, we require* "algorithm-level data attribution" *that measures a data point's general influence on the learning algorithm, independent of specific training random seeds. On the other hand, when interpreting model decisions, we need model-specific data attribution focused on the particular checkpoint we train. Systematically mapping which data attribution approaches best suit specific application scenarios remains an important direction for future research.*

### B.2 WHEN IS RETRAINING-BASED DATA SHAPLEY DESIRABLE FOR DATA ATTRIBUTION?

Retraining-based Data Shapley is still desirable in several scenarios. Firstly, it is more general and applicable to all learning algorithms, whereas In-Run Data Shapley is only applicable to iterative training algorithms. Secondly, retraining-based Shapley is useful when the goal is to understand the contribution of data points to the general learning process rather than to a specific model. Thirdly, because Retraining-based Data Shapley does not require modifying the internal code of the training algorithm, its estimation algorithm, typically Monte Carlo-based, is straightforward and clean to implement.

### B.3 POTENTIAL EXTENSIONS OF IN-RUN DATA SHAPLEY TO OTHER DOMAINS

While this work focuses on data attribution for SGD-trained machine learning models, we believe the core idea of In-Run Data Shapley—decomposing contribution analysis into per-iteration assessments—can potentially be extended to other contexts and applications. Here, we discuss some directions.

**Data attribution for iterative learning algorithms that do not use gradient descent.** Our framework could potentially extend to iterative learning algorithms that don't use gradient descent, such as k-means clustering or decision tree learning. Though these algorithms update models differently, they still proceed iteratively (e.g., k-means alternates between assignment and update steps; decision trees are built through recursive partitioning). For such algorithms, we could analyze how each data point influences these discrete update steps and aggregate these influences across iterations, similar to our approach with gradient-based training.

**Hyperparameter importance.** The framework could potentially be adapted to evaluate hyperparameter contributions during training. One possibility is to set a "baseline" hyperparameter value and assess how choosing a different value impacts each training iteration compared to this baseline. For instance, when evaluating learning rate choices, we could measure how using a specific learning rate value affects model updates compared to using a baseline learning rate. For differentiable hyperparameters, we could leverage Taylor expansion to approximate this difference; for non-differentiable ones, zero-order methods could potentially be used. By accumulating these contributions across training iterations, we could understand hyperparameter impact without requiring multiple complete training runs. At a high level, this view unifies our treatment of training data and hyperparameters - both can be seen as choices that influence each training iteration, where we aim to quantify their impact against baseline scenarios. While technical challenges remain in adapting our method, this direction presents an interesting opportunity for future research.

**Feature attribution (e.g., context-attribution for language models).** Another possible extension we envision is feature attribution for machine learning models. Feature attribution aims at understanding

how each part of the input (like individual words in a sentence or pixels in an image) influences a model's final prediction. When a model makes a prediction, the input features are processed through multiple layers, with each layer transforming the information before passing it to the next layer. Feature attribution aims to track how this information flows and transforms through the network to determine each input feature's contribution to the final output. We envision a *unified theoretical framework* connecting training data attribution and feature attribution, drawing parallels between how information flows during training (through gradient updates) and during prediction (through layer operations). This could lead to more efficient methods for explaining model behavior, particularly for complex architectures like transformers.

## C  MISSING PROOFS

**Theorem 5** (Restate of Theorem 3). *In-Run Data Shapley considering the first-order approximation has closed-form*

$$\phi_z\left(U\right) \approx \sum_{t=0}^{T-1} \phi_z\left(U_{(1)}^{(t)}\right)$$

*where*

$$\phi_z\left(U_{(1)}^{(t)}\right) = -\eta_t \nabla\ell(w_t, z^{(\mathrm{val})}) \cdot \nabla\ell(w_t, z), \quad t = 0, \ldots, T-1$$

*Proof.* For notation simplicity, let $x_j := \nabla\ell(w_t, z_j)$. Given the utility function

$$U_{(1)}^{(t)}(S) = -\eta_t \sum_{z_j \in S} \nabla\ell(w_t, z^{(\mathrm{val})}) \cdot \nabla\ell(w_t, z_j),$$

the marginal contribution of $z$ for any $S \subseteq \mathcal{B}_t \setminus z$ is

$$U_{(2)}^{(t)}(S \cup z) - U_{(2)}^{(t)}(S) = -\eta_t \nabla\ell(w_t, z^{(\mathrm{val})}) \cdot \nabla\ell(w_t, z)$$

Plugging in the expression to the Shapley value's definition immediately gives the result. $\qquad\square$

**Theorem 6** (Restate of Theorem 4). *In-Run Data Shapley considering the second-order approxima-tion has closed-form*

$$\phi_z(U) \approx \sum_{t=0}^{T-1} \left( \phi_z\left(U_{(1)}^{(t)}\right) + \frac{1}{2}\phi_z\left(U_{(2)}^{(t)}\right) \right) \tag{6}$$

*where*

$$\phi_z\left(U_{(1)}^{(t)}\right) + \frac{1}{2}\phi_z\left(U_{(2)}^{(t)}\right) = \underbrace{-\eta_t \nabla \ell(w_t, z^{(\mathrm{val})}) \cdot \nabla \ell(w_t, z)}_{\textcircled{1}\ \textit{influence of } z \textit{ on the loss of } z^{(\mathrm{val})}} + \underbrace{\frac{\eta_t^2}{2} \nabla \ell(w_t, z)^{\mathsf{T}} \boldsymbol{H}_t^{(z^{(\mathrm{val})})} \left( \sum_{z_j \in \mathcal{B}_t} \nabla \ell(w_t, z_j) \right)}_{\textcircled{2}\ \textit{interaction between } z \textit{ and other training points}}$$

$$\tag{7}$$

*for any $t = 0, \ldots, T-1$.*

*Proof.* We show that

$$\phi_z\left(U_{(2)}^{(t)}\right) = \eta_t^2 \nabla \ell(w_t, z) \mathbf{H}_t^{(z^{(\mathrm{val})})} \left( \sum_{z_j \in \mathcal{B}_t} \nabla \ell(w_t, z_j) \right).$$

For notation simplicity, let $x_j := \nabla \ell(w_t, z_j)$, and $\tilde{x} := \nabla \ell(w_t, z)$. Furthermore, let $n_t := |\mathcal{B}_t|$ the batch size in $t$-th iteration. For any $S \subseteq \mathcal{B}_t \setminus z$, the marginal contribution of $z$ is

$$U_{(2)}^{(t)}(S \cup z) - U_{(2)}^{(t)}(S) = \eta_t^2 \left( 2\tilde{x}\mathbf{H}_t^{(z^{(\mathrm{val})})} \left( \sum_{z_j \in S} x_j \right) + \tilde{x}^{\mathsf{T}}\mathbf{H}_t^{(z^{(\mathrm{val})})}\tilde{x} \right).$$

Plug the above expression into the Shapley value's formula, we have

$$\phi_z\left(U_{(2)}^{(t)}\right) = \eta_t^2 \left( \tilde{x}^{\mathsf{T}}\mathbf{H}_t^{(z^{(\mathrm{val})})}\tilde{x} + \frac{2}{n_t}\sum_{k=1}^{n_t}\binom{n_t-1}{k-1}^{-1} \sum_{S \subseteq \mathcal{B}_t \setminus \{z\},\ |S|=k-1} \left[ \tilde{x}^{\mathsf{T}}\mathbf{H}_t^{(z^{(\mathrm{val})})}\sum_{z_j \in S}x_j \right] \right)$$

$$= \eta_t^2 \left( \tilde{x}^{\mathsf{T}}\mathbf{H}_t^{(z^{(\mathrm{val})})}\tilde{x} + \tilde{x}^{\mathsf{T}}\mathbf{H}_t^{(z^{(\mathrm{val})})} \left[ \frac{2}{n_t}\sum_{k=1}^{n_t}\binom{n_t-1}{k-1}^{-1} \sum_{S \subseteq \mathcal{B}_t \setminus \{z\},\ |S|=k-1} \left( \sum_{z_j \in S}x_j \right) \right] \right)$$

$$= \eta_t^2 \left( \tilde{x}^{\mathsf{T}}\mathbf{H}_t^{(z^{(\mathrm{val})})}\tilde{x} + \tilde{x}^{\mathsf{T}}\mathbf{H}_t^{(z^{(\mathrm{val})})} \left[ \frac{2}{n_t}\sum_{k=2}^{n_t}\binom{n_t-1}{k-1}^{-1}\binom{n_t-2}{k-2} \left( \sum_{z_j \in \mathcal{B}_t \setminus \{z\}}x_j \right) \right] \right)$$

$$= \eta_t^2 \left( \tilde{x}^{\mathsf{T}}\mathbf{H}_t^{(z^{(\mathrm{val})})}\tilde{x} + \tilde{x}^{\mathsf{T}}\mathbf{H}_t^{(z^{(\mathrm{val})})} \left[ \frac{2}{n_t}\sum_{k=2}^{n_t}\frac{k-1}{n_t-1} \left( \sum_{z_j \in \mathcal{B}_t \setminus \{z\}}x_j \right) \right] \right)$$

$$= \eta_t^2 \left( \tilde{x}^{\mathsf{T}}\mathbf{H}_t^{(z^{(\mathrm{val})})}\tilde{x} + \tilde{x}^{\mathsf{T}}\mathbf{H}_t^{(z^{(\mathrm{val})})} \left( \sum_{z_j \in \mathcal{B}_t \setminus \{z\}}x_j \right) \right)$$

$$= \eta_t^2 \left[ \tilde{x}^{\mathsf{T}}\mathbf{H}_t^{(z^{(\mathrm{val})})} \left( \sum_{z_j \in \mathcal{B}_t}x_j \right) \right]$$

$$\square$$

## D  TECHNICAL DETAILS

**Notation review.** Consider a linear layer $\mathbf{s} = \mathbf{aW}$, where $\mathbf{W} \in \mathbb{R}^{d_1 \times d_2}$ is the weight matrix, $\mathbf{a} = (\mathbf{a}^{(1)}, \dots, \mathbf{a}^{(B)})^\mathsf{T}$ is the mini-batch input, and $\mathbf{s} = (\mathbf{s}^{(1)}, \dots, \mathbf{s}^{(B)})^\mathsf{T}$ is the output (i.e., the pre-activation tensor). For non-sequential data, $\mathbf{a} \in \mathbb{R}^{B \times d_1}, \mathbf{s} \in \mathbb{R}^{B \times d_2}$. For sequential data with sequence length $T$, $\mathbf{a} \in \mathbb{R}^{B \times d_1 \times T}, \mathbf{s} \in \mathbb{R}^{B \times d_2 \times T}$. Let $\ell^{(i)} := \ell(w, z_i)$ denote the current model's individual loss on $z_i$. For notation convenience, we denote $\mathbf{b}^{(i)} := \frac{\partial \ell^{(i)}}{\partial \mathbf{s}}$.

### D.1  GHOST DOT-PRODUCT

By applying the chain rule, we can express the gradient of an individual loss $\ell^{(i)} := \ell(w, z_i)$ with respect to $\mathbf{W}$ as

$$\frac{\partial \ell^{(i)}}{\partial \mathbf{W}} = \frac{\partial \ell^{(i)}}{\partial \mathbf{s}^{(i)}} \frac{\partial \mathbf{s}^{(i)}}{\partial \mathbf{W}} = \frac{\partial \ell^{(i)}}{\partial \mathbf{s}^{(i)}} \mathbf{a}^{(i)} = \frac{\partial \ell}{\partial \mathbf{s}^{(i)}} \mathbf{a}^{(i)} = \mathbf{a}^{(i)} \otimes \mathbf{b}^{(i)} \tag{8}$$

where $\ell := \sum_{j=1}^{B} \ell^{(j)}$ is the aggregated loss. Note that the individual's output gradient $\mathbf{b}^{(i)} = \frac{\partial \ell^{(i)}}{\partial \mathbf{s}^{(i)}} = \frac{\partial \ell}{\partial \mathbf{s}^{(i)}}$ is readily available during the backpropagation pass.

Suppose we are interested in computing the gradient dot-product $\frac{\partial \ell^{(1)}}{\partial \mathbf{W}} \odot \frac{\partial \ell^{(2)}}{\partial \mathbf{W}}$ between two data points $z_1, z_2$ in the same batch in the backpropagation. We first discuss the case for non-sequential data and then extend it to sequential data.

**Non-sequential data.** For non-sequential data, we have each $\mathbf{a}^{(i)} \in \mathbb{R}^{d_1 \times 1}$ and $\mathbf{b}^{(i)} \in \mathbb{R}^{1 \times d_2}$. By (8), we have

$$\frac{\partial \ell^{(1)}}{\partial \mathbf{W}} \odot \frac{\partial \ell^{(2)}}{\partial \mathbf{W}} = \left( \mathbf{a}^{(1)} \otimes \mathbf{b}^{(1)} \right) \odot \left( \mathbf{a}^{(2)} \otimes \mathbf{b}^{(2)} \right) = \left( \left( \mathbf{b}^{(1)} \right)^\mathsf{T} \left( \mathbf{b}^{(2)} \right) \right) \left( \left( \mathbf{a}^{(1)} \right)^\mathsf{T} \mathbf{a}^{(2)} \right)$$

Hence, we can compute the dot-product between $\frac{\partial \ell^{(1)}}{\partial \mathbf{W}}$ and $\frac{\partial \ell^{(2)}}{\partial \mathbf{W}}$ without actually instantiating the gradient vector $\frac{\partial \ell^{(1)}}{\partial \mathbf{W}}$ or $\frac{\partial \ell^{(2)}}{\partial \mathbf{W}}$. We can take the dot products between $\left( \mathbf{b}^{(1)} \right)^\mathsf{T} \left( \mathbf{b}^{(2)} \right)$ and $\left( \mathbf{a}^{(1)} \right)^\mathsf{T} \mathbf{a}^{(2)}$, and then multiply the results together. Moreover, all of the materials $\mathbf{a}^{(1)}, \mathbf{a}^{(2)}, \mathbf{b}^{(1)}, \mathbf{b}^{(2)}$ that are required for computation are all already available in one backpropagation. Hence, with a single backpropagation, we can efficiently compute the gradient dot-product between every pair of data points within the batch.

**Sequential data.** For sequential data, we have each $\mathbf{a}^{(i)} \in \mathbb{R}^{d_1 \times T}$ and $\mathbf{b}^{(i)} \in \mathbb{R}^{T \times d_2}$. By (8), we have

$$\begin{aligned}
\frac{\partial \ell^{(1)}}{\partial \mathbf{W}} \odot \frac{\partial \ell^{(2)}}{\partial \mathbf{W}} &= \left( \mathbf{a}^{(1)} \otimes \mathbf{b}^{(1)} \right) \odot \left( \mathbf{a}^{(2)} \otimes \mathbf{b}^{(2)} \right) = \sum_{j,k=1}^{d_1,d_2} \left( \mathbf{a}^{(1)} \otimes \mathbf{b}^{(1)} \right)_{j,k} \left( \mathbf{a}^{(2)} \otimes \mathbf{b}^{(2)} \right)_{j,k} \\
&= \sum_{j,k=1}^{d_1,d_2} \left( \mathbf{a}_j^{(1)} \mathbf{b}_k^{(1)} \right) \left( \mathbf{a}_j^{(2)} \mathbf{b}_k^{(2)} \right) \\
&= \sum_{j,k=1}^{d_1,d_2} \left( \sum_{t=1}^{T} \mathbf{a}_{jt}^{(1)} \mathbf{b}_{tk}^{(1)} \right) \left( \sum_{t=1}^{T} \mathbf{a}_{jt}^{(2)} \mathbf{b}_{tk}^{(2)} \right) \\
&= \sum_{t_1,t_2=1}^{T,T} \left( \sum_{j=1}^{d_1} \mathbf{a}_{jt_1}^{(1)} \mathbf{a}_{jt_2}^{(2)} \right) \left( \sum_{k=1}^{d_2} \mathbf{a}_{kt_1}^{(1)} \mathbf{a}_{kt_2}^{(2)} \right) \\
&= \sum_{t_1,t_2=1}^{T,T} \left( \mathbf{a}_{\cdot,t_1}^{(1)} \mathbf{a}_{\cdot,t_2}^{(2)} \right) \left( \mathbf{b}_{\cdot,t_1}^{(1)} \mathbf{b}_{\cdot,t_2}^{(2)} \right) \\
&= \left( \left( \mathbf{b}^{(1)} \right) \left( \mathbf{b}^{(2)} \right)^\mathsf{T} \right) \odot \left( \left( \mathbf{a}^{(1)} \right)^\mathsf{T} \mathbf{a}^{(2)} \right)
\end{aligned}$$

Hence, comparing with directly computing per-sample gradients, if $2T^2 < d_1 d_2$, it is more memory-efficient to first multiply the matrices of $\left(\mathbf{b}^{(1)}\right)\left(\mathbf{b}^{(2)}\right)^{\mathsf{T}}$ and $\left(\mathbf{a}^{(1)}\right)^{\mathsf{T}}\mathbf{a}^{(2)}$, then take the inner product between the two $T \times T$ matrices. If $2T^2 \geq d_1 d_2$, then we can first take the outer products $\mathbf{a}^{(1)} \otimes \mathbf{b}^{(1)}$ and $\mathbf{a}^{(2)} \otimes \mathbf{b}^{(2)}$, then take their inner product. In either case, we only need a single backpropagation to compute the gradient dot product between every pair of data points within the batch, similar to the case of non-sequential data.

**Remark 5** (Applications of "Ghost" Technique Beyond Data Attribution). *The techniques developed in this work for efficiently computing gradient dot products may be of independent interest, as this operation arises in various machine learning algorithms and applications beyond data attribution. For instance, the "ghost dot-product" technique can be used in calculating the cosine similarity between individual gradients (together with the "ghost clipping" technique), which is frequently being employed to analyze training dynamics (Fort & Ganguli, 2019) and detect adversarial attacks (Dhaliwal & Shintre, 2018). Additionally, gradient similarity is used in active learning to select the most informative data points for labeling (Ash et al., 2019) and in multitask learning to balance the contributions of different tasks (Yu et al., 2020).*

### D.2 GHOST GRADIENT-HESSIAN-GRADIENT PRODUCT

We denote $L$ as the number of layers in a neural network, and we denote $\mathbf{W}_i$, $i = 1, \ldots, L$ as the weight parameters of each layer. We denote $\mathbf{H}_{\mathbf{W}_i}^{(z^{(\text{val})})}$ as the Hessian matrix of $\ell^{(z^{(\text{val})})}$ on layer $\mathbf{W}_i$. Suppose we are interested in computing the gradient dot product $\nabla\ell^{(1)}\mathbf{H}^{(z^{(\text{val})})}\nabla\ell^{(2)}$.

$$\nabla\ell^{(1)}\mathbf{H}^{(z^{(\text{val})})}\nabla\ell^{(2)} = \sum_{i,j=1}^{L,L} \frac{\partial\ell^{(1)}}{\partial\mathbf{W}_i} \left( \frac{\partial^{(2)}\ell^{(z^{(\text{val})})}}{\partial\mathbf{W}_i\partial\mathbf{W}_j} \right) \frac{\partial\ell^{(2)}}{\partial\mathbf{W}_j}$$

$$= \sum_{i,j=1}^{L,L} (\mathbf{a}_i^{(1)} \otimes \mathbf{b}_i^{(1)}) \left( \frac{\partial^{(2)}\ell^{(z^{(\text{val})})}}{\partial\mathbf{W}_i\partial\mathbf{W}_j} \right) (\mathbf{a}_j^{(2)} \otimes \mathbf{b}_j^{(2)})$$

We first look at the Hessian-vector product.

$$\left( \frac{\partial^{(2)}\ell^{(z^{(\text{val})})}}{\partial\mathbf{W}_i\partial\mathbf{W}_j} \right) (\mathbf{a}_j^{(2)} \otimes \mathbf{b}_j^{(2)}) = \frac{\partial}{\partial\mathbf{W}_i} \left[ \frac{\partial\ell^{(z^{(\text{val})})}}{\partial\mathbf{W}_j}(\mathbf{a}_j^{(2)} \otimes \mathbf{b}_j^{(2)}) \right]$$

$$= \frac{\partial}{\partial\mathbf{W}_i} \left[ (\mathbf{a}_j^{(z^{(\text{val})})} \otimes \mathbf{b}_j^{(z^{(\text{val})})})(\mathbf{a}_j^{(2)} \otimes \mathbf{b}_j^{(2)}) \right]$$

$$= \frac{\partial}{\partial\mathbf{W}_i} \left[ (\mathbf{a}_j^{(z^{(\text{val})})}{}'\mathbf{a}_j^{(2)})(\mathbf{b}_j^{(z^{(\text{val})})}{}'\mathbf{b}_j^{(2)}) \right]$$

$$= \frac{\partial}{\partial\mathbf{W}_i} [\mathbf{c}_j\mathbf{d}_j]$$

$$= \mathbf{c}_j \frac{\partial\mathbf{d}_j}{\partial\mathbf{W}_i} + \mathbf{d}_j \frac{\partial\mathbf{c}_j}{\partial\mathbf{W}_i}$$

where we denote $\mathbf{c}_j := \mathbf{a}_j^{(z^{(\text{val})})}{}'\mathbf{a}_j^{(2)}$ and $\mathbf{d}_j := \mathbf{b}_j^{(z^{(\text{val})})}{}'\mathbf{b}_j^{(2)}$.

We run the second backpropagation on the gradient dot product $\nabla\ell^{(z^{(\text{val})})}\nabla\ell^{(2)} = \sum_{j=1}^{L} \mathbf{c}_j\mathbf{d}_j$. Note that here $\mathbf{a}_j^{(2)}, \mathbf{b}_j^{(2)}$ are constant vectors. For $\frac{\partial\mathbf{c}_j}{\partial\mathbf{W}_i}$, we have the similar decomposition

$$\frac{\partial\mathbf{c}_j}{\partial\mathbf{W}_i} = \frac{\partial}{\partial\mathbf{W}_i} \left( \mathbf{a}_j^{(z^{(\text{val})})}{}'\mathbf{a}_j^{(2)} \right)$$

$$= \frac{\partial}{\partial\mathbf{s}_i^{(z^{(\text{val})})}} \left( \mathbf{a}_j^{(z^{(\text{val})})}{}'\mathbf{a}_j^{(2)} \right) \otimes \mathbf{a}_i^{(z^{(\text{val})})}$$

For $\frac{\partial\mathbf{d}_j}{\partial\mathbf{W}_i}$, the derivative is more complicated as the $\mathbf{b}_j^{(z^{(\text{val})})}$ also depends on the output from deeper layers. When $j \geq i$, we still have

$$\frac{\partial\mathbf{d}_j}{\partial\mathbf{W}_i} = \frac{\partial}{\partial\mathbf{s}_i^{(z^{(\text{val})})}} \left( \mathbf{b}_j^{(z^{(\text{val})})}{}'\mathbf{b}_j^{(2)} \right) \otimes \mathbf{a}_i^{(z^{(\text{val})})}$$

When $j < i$, we have

$$
\begin{aligned}
\frac{\partial \mathbf{d}_j}{\partial \mathbf{W}_i} &= \frac{\partial \mathbf{b}_j^{(z^{(\text{val})})} \mathbf{b}_j^{(2)}}{\partial \mathbf{W}_i} \\
&= \frac{\partial}{\partial \mathbf{W}_i} \left( \mathbf{b}_j^{(z^{(\text{val})})} \mathbf{b}_j^{(2)} \right) \\
&= \frac{\partial}{\partial \mathbf{W}_i} \left( \frac{\partial \ell^{(z^{(\text{val})})}}{\partial \mathbf{s}_i^{(z^{(\text{val})})}} \frac{\partial \mathbf{s}_i^{(z^{(\text{val})})}}{\partial \mathbf{a}_i^{(z^{(\text{val})})}} \frac{\partial \mathbf{a}_i^{(z^{(\text{val})})}}{\partial \mathbf{s}_{i-1}^{(z^{(\text{val})})}} \cdots \frac{\partial \mathbf{a}_j^{(z^{(\text{val})})}}{\partial \mathbf{s}_j^{(z^{(\text{val})})}} \mathbf{b}_j^{(2)} \right) \\
&= \frac{\partial}{\partial \mathbf{W}_i} \left( \frac{\partial \ell^{(z^{(\text{val})})}}{\partial \mathbf{s}_i^{(z^{(\text{val})})}} \mathbf{W}_i \frac{\partial \mathbf{a}_i^{(z^{(\text{val})})}}{\partial \mathbf{s}_{i-1}^{(z^{(\text{val})})}} \cdots \frac{\partial \mathbf{a}_j^{(z^{(\text{val})})}}{\partial \mathbf{s}_j^{(z^{(\text{val})})}} \mathbf{b}_j^{(2)} \right) \\
&= \frac{\partial}{\partial \mathbf{s}_i^{(z^{(\text{val})})}} \left( \frac{\partial \ell^{(z^{(\text{val})})}}{\partial \mathbf{s}_i^{(z^{(\text{val})})}} \mathbf{W}_i \frac{\partial \mathbf{a}_i^{(z^{(\text{val})})}}{\partial \mathbf{s}_{i-1}^{(z^{(\text{val})})}} \cdots \frac{\partial \mathbf{a}_j^{(z^{(\text{val})})}}{\partial \mathbf{s}_j^{(z^{(\text{val})})}} \mathbf{b}_j^{(2)} \right) \otimes \mathbf{a}_i^{(z^{(\text{val})})} \\
&\quad + \frac{\partial \ell^{(z^{(\text{val})})}}{\partial \mathbf{s}_i^{(z^{(\text{val})})}} \otimes \left( \frac{\partial \mathbf{a}_i^{(z^{(\text{val})})}}{\partial \mathbf{s}_{i-1}^{(z^{(\text{val})})}} \cdots \frac{\partial \mathbf{a}_j^{(z^{(\text{val})})}}{\partial \mathbf{s}_j^{(z^{(\text{val})})}} \mathbf{b}_j^{(2)} \right) \\
&= \frac{\partial}{\partial \mathbf{s}_i^{(z^{(\text{val})})}} (\mathbf{b}_j^{(z^{(\text{val})})} \mathbf{b}_j^{(2)}) \otimes \mathbf{a}_i^{(z^{(\text{val})})} + \mathbf{b}_i^{(z^{(\text{val})})} \otimes \left( \frac{\partial \mathbf{a}_i^{(z^{(\text{val})})}}{\partial \mathbf{s}_j^{(z^{(\text{val})})}} \mathbf{b}_j^{(2)} \right)
\end{aligned}
$$

Overall,

$$
\begin{aligned}
\frac{\partial \mathbf{d}_j}{\partial \mathbf{W}_i} &= \frac{\partial}{\partial \mathbf{W}_i} (\mathbf{b}_j^{(z^{(\text{val})})} \mathbf{b}_j^{(2)}) \\
&= \begin{cases} \frac{\partial}{\partial \mathbf{s}_i^{(z^{(\text{val})})}} \left( \mathbf{b}_j^{(z^{(\text{val})})} \mathbf{b}_j^{(2)} \right) \otimes \mathbf{a}_i^{(z^{(\text{val})})} & j \geq i \\ \frac{\partial}{\partial \mathbf{s}_i^{(z^{(\text{val})})}} \left( \mathbf{b}_j^{(z^{(\text{val})})} \mathbf{b}_j^{(2)} \right) \otimes \mathbf{a}_i^{(z^{(\text{val})})} + \mathbf{b}_i^{(z^{(\text{val})})} \otimes \left( \frac{\partial \mathbf{a}_i^{(z^{(\text{val})})}}{\partial \mathbf{s}_j^{(z^{(\text{val})})}} \mathbf{b}_j^{(2)} \right) & j < i \end{cases}
\end{aligned}
$$

Hence, we have

$$
\nabla \ell^{(1)} \mathbf{H}^{(z^{(\text{val})})} \nabla \ell^{(2)}
$$
$$
= \sum_{i,j=1}^{L,L} (\mathbf{a}_i^{(1)} \otimes \mathbf{b}_i^{(1)}) \left[ \mathbf{c}_j \frac{\partial \mathbf{d}_j}{\partial \mathbf{W}_i} + \mathbf{d}_j \frac{\partial \mathbf{c}_j}{\partial \mathbf{W}_i} \right]
$$
$$
= \underbrace{\sum_{i,j=1}^{L,L} (\mathbf{a}_i^{(1)} \otimes \mathbf{b}_i^{(1)}) \left( \frac{\partial}{\partial \mathbf{s}_i^{(z^{(\text{val})})}} (\mathbf{c}_j \mathbf{d}_j) \otimes \mathbf{a}_i^{(z^{(\text{val})})} \right)}_{\textcircled{1}} + \underbrace{\sum_{i>j}^{L,L} (\mathbf{a}_i^{(1)} \otimes \mathbf{b}_i^{(1)}) \left( \mathbf{b}_i^{(z^{(\text{val})})} \otimes \left( \frac{\partial \mathbf{a}_i^{(z^{(\text{val})})}}{\partial \mathbf{s}_j^{(z^{(\text{val})})}} \mathbf{b}_j^{(2)} \right) \right)}_{\textcircled{2}}
$$

For the first part, we have

$$
\begin{aligned}
\textcircled{1} &= \sum_{i=1}^{L} (\mathbf{a}_i^{(1)} \otimes \mathbf{b}_i^{(1)}) \left( \frac{\partial}{\partial \mathbf{s}_i^{(z^{(\text{val})})}} \left( \sum_{j=1}^{L} \mathbf{c}_j \mathbf{d}_j \right) \otimes \mathbf{a}_i^{(z^{(\text{val})})} \right) \\
&= \sum_{i=1}^{L} \left( \mathbf{a}_i^{(1)} \mathbf{a}_i^{(z^{(\text{val})})} \right) \left( \mathbf{b}_i^{(1)} \left( \frac{\partial}{\partial \mathbf{s}_i^{(z^{(\text{val})})}} \left( \sum_{j=1}^{L} \mathbf{c}_j \mathbf{d}_j \right) \right) \right)
\end{aligned}
$$

where $\frac{\partial}{\partial \mathbf{s}_i^{(z^{(\text{val})})}} \left( \sum_{j=1}^{L} \mathbf{c}_j \mathbf{d}_j \right)$, $i = 1, \ldots, L$ is readily available from the second backpropagation.

For the second part, we have

$$\textcircled{2} = \sum_{i>j}^{L,L} (\mathbf{a}_i^{(1)} \otimes \mathbf{b}_i^{(1)}) \left( \mathbf{b}_i^{(z^{(\mathrm{val})})} \otimes \left( \frac{\partial \mathbf{a}_i^{(z^{(\mathrm{val})})}}{\partial \mathbf{s}_j^{(z^{(\mathrm{val})})}} \mathbf{b}_j^{(2)} \right) \right)$$

$$= \sum_{i>j}^{L,L} \left( \mathbf{a}_i^{(1)} \left( \frac{\partial \mathbf{a}_i^{(z^{(\mathrm{val})})}}{\partial \mathbf{s}_j^{(z^{(\mathrm{val})})}} \mathbf{b}_j^{(2)} \right) \right) \left( \mathbf{b}_i^{(1)} \mathbf{b}_i^{(z^{(\mathrm{val})})} \right)$$

The remaining question is how to compute $f(i,j) := \mathbf{a}_i^{(1)} \frac{\partial \mathbf{a}_i^{(z^{(\mathrm{val})})}}{\partial \mathbf{s}_j^{(z^{(\mathrm{val})})}} \mathbf{b}_j^{(2)}$. Observe that

$$\mathbf{a}_i^{(1)} \frac{\partial \mathbf{a}_i^{(z^{(\mathrm{val})})}}{\partial \mathbf{s}_j^{(z^{(\mathrm{val})})}} \mathbf{b}_j^{(2)} = \mathbf{a}_i^{(1)} \frac{\partial \mathbf{a}_i^{(z^{(\mathrm{val})})}}{\partial \mathbf{s}_{i-1}^{(z^{(\mathrm{val})})}} \frac{\partial \mathbf{s}_{i-1}^{(z^{(\mathrm{val})})}}{\partial \mathbf{a}_{i-1}^{(z^{(\mathrm{val})})}} \cdots \frac{\partial \mathbf{s}_{j+1}^{(z^{(\mathrm{val})})}}{\partial \mathbf{a}_{j+1}^{(z^{(\mathrm{val})})}} \frac{\partial \mathbf{a}_{j+1}^{(z^{(\mathrm{val})})}}{\partial \mathbf{s}_j^{(z^{(\mathrm{val})})}} \mathbf{b}_j^{(2)}$$

$$= \mathbf{a}_i^{(1)} \frac{\partial \mathbf{a}_i^{(z^{(\mathrm{val})})}}{\partial \mathbf{s}_{i-1}^{(z^{(\mathrm{val})})}} \mathbf{W}_{i-1} \cdots \mathbf{W}_{j+1} \frac{\partial \mathbf{a}_{j+1}^{(z^{(\mathrm{val})})}}{\partial \mathbf{s}_j^{(z^{(\mathrm{val})})}} \mathbf{b}_j^{(2)}$$

Moreover, $\frac{\partial \mathbf{a}_{j+1}^{(z^{(\mathrm{val})})}}{\partial \mathbf{s}_j^{(z^{(\mathrm{val})})}}$ is easy to compute as the only operation between $\mathbf{a}_{j+1}^{(z^{(\mathrm{val})})}$ and $\mathbf{s}_j^{(z^{(\mathrm{val})})}$ is an element-wise activation function. For example, if the activation function is ReLU, then $\frac{\partial \mathbf{a}_{j+1}^{(z^{(\mathrm{val})})}}{\partial \mathbf{s}_j^{(z^{(\mathrm{val})})}}$ will simply be a matrix that contains elements in $\{0,1\}$, depending on the sign of the position in $\mathbf{s}_j^{(z^{(\mathrm{val})})}$. To efficiently compute $f(i,j)$ for all $i > j$, for each $j = 1, \ldots, L$, we can maintain the quantity

$$\mathbf{g}(i,j) := \frac{\partial \mathbf{a}_i^{(z^{(\mathrm{val})})}}{\partial \mathbf{s}_{i-1}^{(z^{(\mathrm{val})})}} \mathbf{W}_{i-1} \cdots \mathbf{W}_{j+1} \frac{\partial \mathbf{a}_{j+1}^{(z^{(\mathrm{val})})}}{\partial \mathbf{s}_j^{(z^{(\mathrm{val})})}} \mathbf{b}_j^{(2)}$$

and we have $f(i,j) = \mathbf{a}_i^{(1)} \mathbf{g}(i,j)$. To compute $f(i+1,j)$, note that $\mathbf{g}(i+1,j) = \frac{\partial \mathbf{a}_{i+1}^{(z^{(\mathrm{val})})}}{\partial \mathbf{s}_i^{(z^{(\mathrm{val})})}} \mathbf{W}_i \mathbf{g}(i,j)$. Moreover, we note that in our scenario, we are interested in the gradient dot-product

$$\nabla \ell^{(1)} \mathbf{H}^{(z^{(\mathrm{val})})} \left( \sum_{m=1}^{B} \nabla \ell^{(m)} \right)$$

where the Hessian-vector product $\mathbf{H}^{(z^{(\mathrm{val})})} \left( \sum_{m=1}^{B} \nabla \ell^{(m)} \right)$ is fixed for any training point corresponds to $\ell^{(1)}$. Therefore, the second backpropagation only needs to be taken once as the gradient vector on the right-hand side is always fixed.

### D.3 Merging Batch Selection and Gradient Update in One Backpropagation

By utilizing the ghost Dot-Product technique developed in this paper, we can calculate or approximate all importance scores and correction terms in a single backpropagation pass, without materializing any model-sized vectors. To compute the gradient dot-product between each training point $z_i \in \mathcal{B}_t$ and the validation data $z^{(\mathrm{val})}$, we propose including $z^{(\mathrm{val})}$ in the backpropagation along with the training batch. Specifically, we can backpropagate with respect to $\left( \sum_{z_i \in \mathcal{B}_t} \ell^{(i)} \right) + \ell^{(z^{(\mathrm{val})})}$. After computing the In-Run Data Shapley scores for the current iteration, it may seem necessary to backpropagate with respect to $\sum_{z_i \in \mathcal{B}_t} \ell^{(i)}$ to compute the gradient for the parameter update. However, this is not required. We can simply reuse the output gradient $\frac{\partial \ell^{(i)}}{\partial \mathbf{s}^{(i)}}$ from the original backpropagation and aggregate the gradients for all selected data points. This technique is adapted from the "book-keeping trick" proposed in Bu et al. (2023).

# E EVALUATION

## E.1 DATA PREPROCESSING & TRAINING SETTINGS & IMPLEMENTATION DETAILS

We conduct the pretraining experiment using the GPT2 (124M) model (Radford et al., 2019) and Pythia-410M (Biderman et al., 2023) on the Pile dataset. Our codebase is adapted from `https://github.com/karpathy/nanoGPT/tree/master`. We first tokenize and split the entire dataset into chunks, and store them in the disk in numpy array format, which significantly speeds up data loading. The maximum sequence length is set to 1024. The learning rate is set at a maximum of 0.0006, with a minimum learning rate of 0.00006. We use AdamW as the optimizer with a weight decay of 0.1, and beta values set to 0.9 and 0.95. Gradients are clipped at a maximum value of 1.0 to maintain stability during training. The batch size is set to 16, with a learning rate warmup of 2000 iterations. Due to the shortage of computation resources, we stop the training at 500,000 iterations.

**Methods for comparison.** In Section 5, we compare In-Run Data Shapley with the influence function (Koh & Liang, 2017), which approximates the change in the model's loss on the test example when the training example is removed from the training set (i.e., the leave-one-out score). BM25 (Robertson et al., 2009) featurizes examples by their word frequency statistics to rank the training instances. We measure the similarity between the query examples and the training corpus, using the similarity scores as the value scores for the training corpus. we set the hyperparameters to $k_1 = 1.5, b = 0.75$, which are commonly used values that balance the term frequency scaling and document length normalization.

**Remark 6.** *We do not compare with TRAK (Park et al., 2023) as it generally requires aggregating the estimator across multiple trained models for a reasonable performance, and the original codebase is not applicable to Huggingface models. In their original paper, the largest scale experiment is fine-tuning BERT-base.*

## E.2 ADDITIONAL FIDELITY EVALUATION

### E.2.1 APPROXIMATION ERROR FOR IN-RUN DATA SHAPLEY WITH LARGER LEARNING RATES

To further investigate the impact of learning rate on the approximation quality of In-Run Data Shapley, we conducted additional experiments to Section 5.2 with larger learning rates of $6 \times 10^{-4}$ and $5 \times 10^{-3}$, where the latter being significantly higher than typical learning rates used in foundation model pretraining. Figure 6 shows the comparison between Monte Carlo-estimated In-Run Data Shapley and our approximations with learning rate $\eta = 6 \times 10^{-4}$. Both first- and second-order approximations achieve remarkably high accuracy with RMSE = 0.0012 and Spearman correlation = 0.99. Even with an extremely large learning rate of $\eta = 5 \times 10^{-3}$ (Figure 7), our approximations maintain reasonable accuracy with RMSE $\approx 0.0017$ and Spearman correlation = 0.79. The strong rank correlation, particularly at learning rates typical for model pretraining ($\eta \leq 10^{-3}$), suggests that our approximations effectively preserve the relative importance of training data points. This property is crucial for practical applications such as identifying low-quality data or selecting valuable training examples, where relative rankings are more important than absolute values.

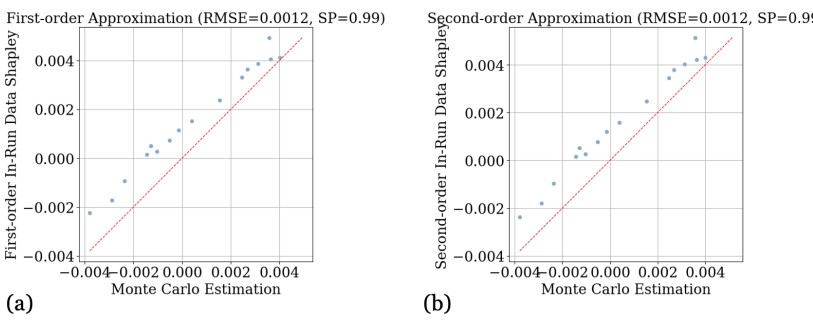

Figure 6: Comparison between the Monte Carlo-estimated In-Run Data Shapley and First/Second-order In-Run Data Shapley (learning rate = $6 \times 10^{-4}$).

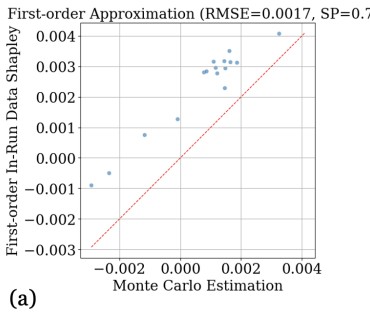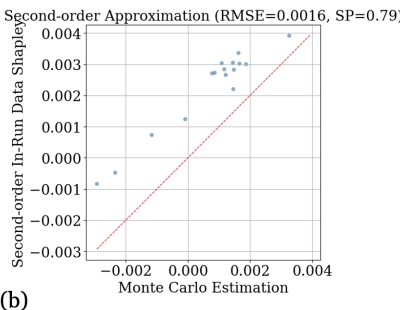

Figure 7: Comparison between the Monte Carlo-estimated In-Run Data Shapley and First/Second-order In-Run Data Shapley (learning rate $= 5 \times 10^{-3}$).

### E.2.2 APPROXIMATION ERROR FOR UTILITY FUNCTION

In this experiment, we empirically investigate the errors of the first- and second-order approximations to $U^{(t)}$ on GPT2, as proposed in Section 4.1. The correlations are shown in Figure 8. As we can see, even the first-order approximation achieves a high approximation accuracy, and the inclusion of the second-order interaction term does not provide a notable improvement in accuracy.

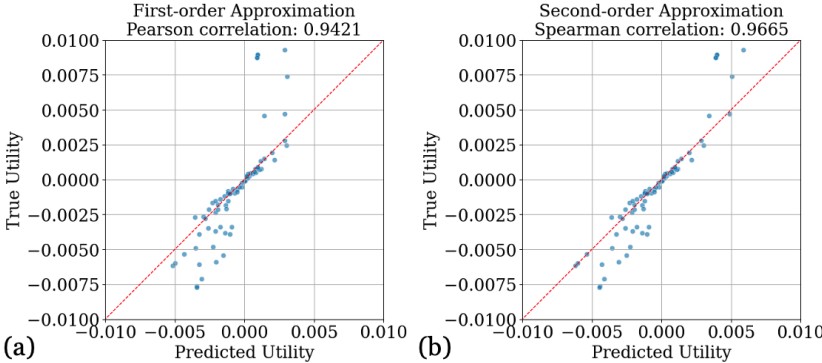

Figure 8: **(a)** We show the correlation between the ground-truth model validation loss change in one gradient update iteration $U^{(t)}(S; z^{(\mathrm{val})}) := \ell(\widetilde{w}_{t+1}(S), z^{(\mathrm{val})}) - \ell(w_t, z^{(\mathrm{val})})$ and the first-order Taylor approximation. **(b)** We show the correlation between $U^{(t)}(S; z^{(\mathrm{val})})$ and the second-order approximation.

**Remark 7** (Difficulty in Extending to Adam). *The techniques developed in this section are specifically tailored for SGD. It is not directly extendable to other popular optimizers like Adam due to their normalization terms. Nonetheless, using SGD as a proxy for Adam allows for efficient data attribution, which is the approach we adopted in practice and has proved to be effective in our experiment. This provides a practical and effective solution for the current scope of our work. Extending these techniques to support Adam and similar optimizers remains an exciting direction for future research.*

### E.3 DETAILS ABOUT SECTION 5.3.1

In this experiment, we first conduct one training run for 20,000 iterations. Among all the corpus that has been used in the training, we compute their In-Run Data Shapley and filter out all corpus among this subset that has negative contributions to model training. After filtering out the 16% negative valued corpus, we train another model on the remaining dataset for 10,000 iterations with all hyperparameters staying the same.

Despite the Pile dataset undergoing multiple layers of data curation (Gao et al., 2020), we still find a significant amount of poor-quality corpus through In-Run Data Shapley. For example, we discovered a large amount of corpora in the PubMed Central domain that are simply empty spaces. Figure 9 provides two examples of the corpus that received the lowest In-Run Data Shapley scores. Additionally, we found several corpora with negative values that appear normal, which is likely due to

a distribution shift with the validation corpus. Crafting a representative validation set is an important direction for future work.

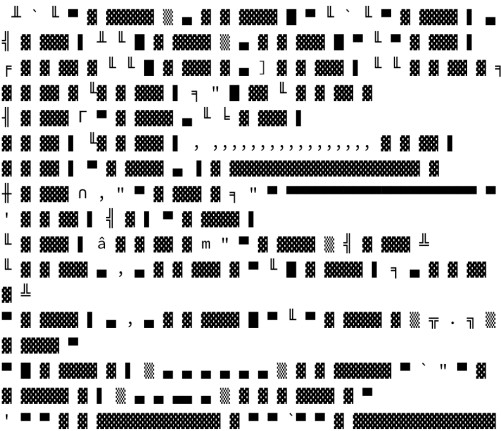

| Corpus with meaningless numbers |
|---|
| ,2,3,3,1,3,0,3,0,2,0,0,      0,0,0,0,0,0,0,0,0,0,2,0... |
| 3,3,3,3,3,3,3,3,1,3,3,...    0,0,0,0,0,0,0,0,0,0,2,... |
| 3,3,3,3,3,3,3,3,3,0,3,...    1,0,0,0,0,0,0,0,0,0,2,... |
| 3,3,3,3,3,3,3,2,3,3,1,...    0,0,0,0,0,0,0,0,0,0,3,... |
| 3,3,3,3,3,1,3,3,3,3,3,... 0,0,0,0,0,0,0,0,0,0,1,... |

Figure 9: Examples of low-quality corpus in Pile found by In-Run Data Shapley. Left: a corpus with meaningless numbers. Right: a corpus with meaningless symbols (likely due to errors from web crawling.)

### E.3.1 Data Selection Performance on Pythia 410M

Figure 10 shows a performance comparison between the original training run and the model trained on the cleaned subsets on Pythia-410M. Similar to the results on GPT2, we can see that removing lower-quality data leads to a significantly faster drop in test loss compared to the original training run, highlighting the effectiveness of In-Run Data Shapley across different architectures.

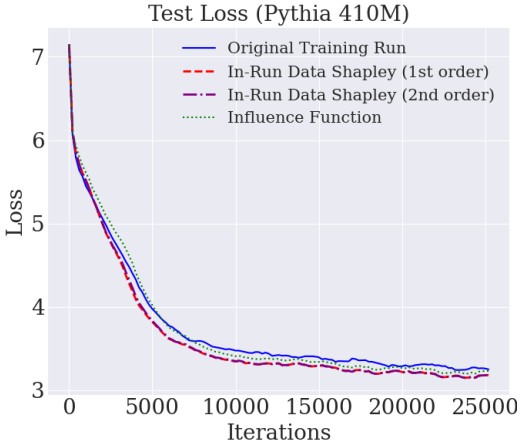

Figure 10: Test loss comparison between the original training run and model trained on the cleaned subset according to different data attribution techniques for Pythia-410M.

### E.4 Additional Results for Section 5.3.2

In addition to Figure 3, we also plot the average value scores of corpora from each domain, averaged across all corpora within the same domain that has been used in gradient updates, over the first 10,000 gradient update iterations. As training progresses, the magnitude of data value scores converges towards zero, which aligns with the diminishing returns property of neural network training. This indicates that the contribution of data points depends on the order in which they are used during training, an aspect that Retraining-based Data Shapley methods cannot capture. Data points introduced in later stages of training contribute less to model performance improvement.

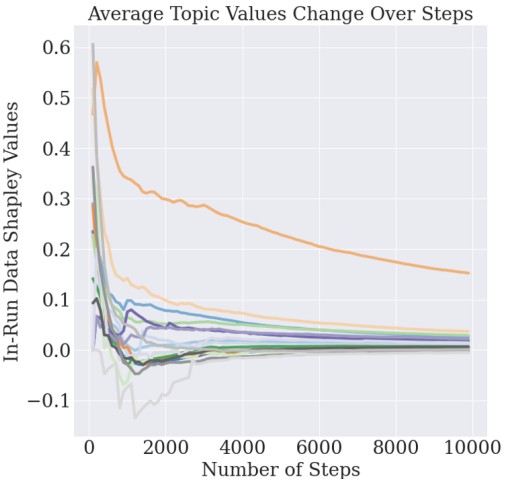

Figure 11: The change of average contribution for data points from each domain to a corpus of math text (same setting as Figure 3).

### E.5 DETAILS ABOUT SECTION 5.3.3

In Section 5.3.3, we use GPT-4 to generate varying levels of paraphrased versions of an original corpus from the training set, categorized as "Partial exactly the same," "Paraphrase," "Significant paraphrase," and "Similar topic." For "Partial exactly the same," the corpus is used as the answer in a creative writing instruction-answer pair. The specific prompt for prompting GPT-4 with examples of the full dialogue are shown in Figure 13, 14, 15. We generate 3 sets of corpus groups, and the experiment results are taken as the average. Figure 12 shows the distribution of calculated In-Run Data Shapley. As we can see, most corpora's values are around 0, but certain corpora have significantly higher values than the other, which means their contributions are higher for the target validation corpus.

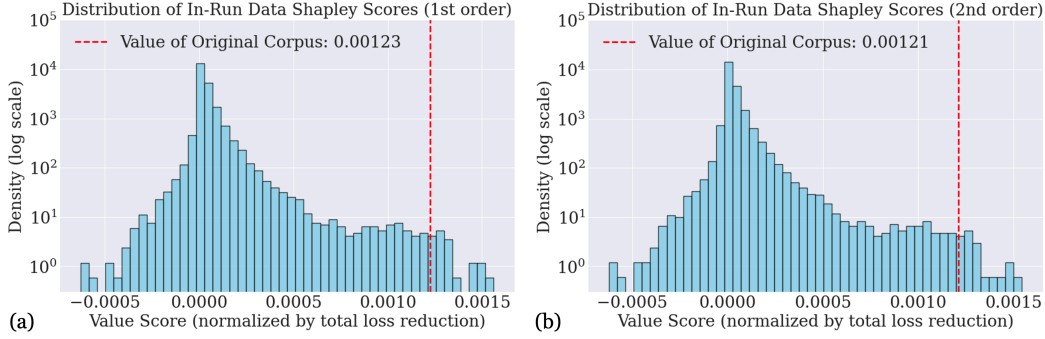

Figure 12: The distribution of calculated In-Run Data Shapley values.

**Dialogue with GPT4 for generating corpus of varying levels of similarity (Original corpus from Wikipedia)**

▶*[Generate "Partial exactly the same"]* <Human> Instruction: [INSERT]
Answer: In 2012, Radhi recruited new 'musicians' for OAG, who were selected from among the students of Akademi Seni Budaya dan Warisan Kebangsaan (). The new line-up consists ofe Qi Razali (drums/backing vocals - original drummer back to the drum position), Muhamad Nizam (guitar - since 2005), Nazrin Zabidi (bass guitar/backing vocals), and Izmer Khasbullah (keyboard/backing vocals). Radhi defines OAG's new music as aslinatif, a combination of muzik asli or traditional music and alternative. Shortly after their reunion, they released the soundtrack and theme song to the TV series, Friday I'm in Love. Few other singles like Si Pengasih Muda, Di Bawah Tanah Ayer and Ambisi" are set to be released soon and also a full studio album coming afterwards.
Please help filling in [INSERT] as the instruction for a creative writing task.

<GPT4> Write a short story about Radhi and his band OAG as they recruit new members and blend traditional music with alternative sounds. Describe their creative challenges and successes leading up to their new releases.

▶*[Generate "Paraphrase"]* <Human> Could you paraphrase the paragraph start from "In 2012, Radhi recruited ..."?

<GPT4> In 2012, Radhi assembled new 'musicians' for OAG, choosing from students at the Akademi Seni Budaya dan Warisan Kebangsaan. The updated lineup includes Qi Razali on drums and backing vocals, returning to his original role, along with Muhamad Nizam on guitar (since 2005), Nazrin Zabidi playing bass and providing backing vocals, and Izmer Khasbullah on keyboards and backing vocals. Radhi describes OAG's revamped sound as 'aslinatif,' a fusion of traditional (muzik asli) and alternative music. Following their reformation, they quickly launched the soundtrack and theme for the TV series "Friday I'm in Love." Upcoming releases include singles like "Si Pengasih Muda," "Di Bawah Tanah Ayer," and "Ambisi," along with a new studio album.

▶*[Generate "Significant paraphrase"]* <Human> Could you make it less look like the original paragraph? Feel free to modify the sentences.

<GPT4> In 2012, Radhi selected a new set of musicians from the students of Akademi Seni Budaya dan Warisan Kebangsaan to join OAG. The band's updated composition includes Qi Razali, who returned to his original role on drums and backing vocals, alongside Muhamad Nizam who has been on guitar since 2005, Nazrin Zabidi on bass with backing vocals, and Izmer Khasbullah playing the keyboard and adding backing vocals. Radhi has termed the band's evolved musical expression as 'aslinatif,' a blend of traditional and alternative sounds. Following their comeback, their first major project was producing the soundtrack and theme for the TV series "Friday I'm in Love." The band is also planning to launch several singles such as "Si Pengasih Muda," "Di Bawah Tanah Ayer," and "Ambisi," and is working towards a new full-length album.

▶*[Generate "Similar topic"]* <Human> (COPY PREVIOUS INSTRUCTION-ANSWER PAIR) ... The above is an example of instruction-answer pair of creative writing. Could you write an instruction-answer pair regarding a less similar but not completely irrelevant topic?

<GPT4> ### Instruction: Write a short story about a classical violinist who decides to explore jazz music, detailing her first encounter with a jazz band. ### Answer: Elena, a classically trained violinist known for her precise and emotive performances, felt something was missing in her musical life. Curiosity led her to a dimly lit jazz club downtown, where the improvised rhythms and melodies captivated her. Hesitantly, she approached the band after their set, violin in hand. They invited her to join them for a jam session the following week. As she stepped onto the stage, her traditional techniques blended unexpectedly with the saxophonist's smooth runs and the drummer's rhythmic beats. The fusion of classical precision and jazz spontaneity created a thrilling new sound. Elena found herself reinvigorated, her violin singing with a freedom she had never before experienced.

Figure 13: Our dialogue with GPT4 for generating corpus of varying levels of similarity for the original corpus from Wikipedia.

**Example Dialogue with GPT4 for generating corpus of varying levels of similarity (Original corpus from CNN)**

▶*[Generate "Partial exactly the same"]* <Human> Instruction: [INSERT]
Answer: The morning after Obama's win, The New York Times declared "Racial Barrier Falls in Decisive Victory." The Los Angeles Times asserted that "for the first time in human history, a largely white nation has elected a black man to be its paramount leader." Some black commentators openly wept on election night, thrilled with witnessing the election of our first black president. Even Sen. John McCain, conceding defeat, pointed to the greatness of the American promise fulfilled in the election of his opponent. Obama's victory offered the possibility that the scars of America's racial legacy were healed or, at least, that they were less raw.
Please help filling in [INSERT] as the instruction for a creative writing task.

<GPT4> Write a reflective essay on the significance of Barack Obama's 2008 presidential election victory, focusing on its impact on race relations in the United States and the emotional responses it elicited from different communities.

▶*[Generate "Paraphrase"]* <Human> Could you paraphrase the paragraph?

<GPT4> The morning after Obama's victory, The New York Times proclaimed "Racial Barrier Falls in Decisive Victory," while The Los Angeles Times noted that "for the first time in human history, a predominantly white nation has elected a black man as its paramount leader." Some black commentators wept openly on election night, overjoyed at witnessing the election of the first black president. Even Sen. John McCain, in his concession speech, highlighted the fulfillment of the American promise through his opponent's win. Obama's triumph suggested the possibility that the deep wounds of America's racial history were healing or, at the very least, becoming less acute.

▶*[Generate "Significant paraphrase"]* <Human> Could you make it less look like the original paragraph? Feel free to modify the sentences.

<GPT4> The morning after Obama's victory, headlines captured the historic moment: The New York Times announced "Racial Barrier Falls in Decisive Victory," and The Los Angeles Times highlighted that "for the first time, a predominantly white nation has chosen a black man as its leader." On election night, many black commentators shed tears of joy, moved by the milestone of electing the first black president. Even Sen. John McCain, in his concession speech, acknowledged the realization of the American promise in his opponent's success. Obama's win signaled a potential healing of America's racial wounds or at least a step toward easing their pain.

▶*[Generate "Similar topic"]* <Human> (COPY PREVIOUS INSTRUCTION-ANSWER PAIR) ... The above is an example of instruction-answer pair of creative writing. Could you write an instruction-answer pair regarding a less similar but not completely irrelevant topic?

<GPT4> ### Instruction: Write a reflective essay on the significance of the Supreme Court's 2015 decision to legalize same-sex marriage nationwide, focusing on its impact on LGBTQ+ rights in the United States and the emotional responses it elicited from different communities.
### Answer: The morning after the Supreme Court's landmark decision, The New York Times declared "Equal Dignity in the Eyes of the Law." The Washington Post highlighted, "Love Wins: Supreme Court Affirms Same-Sex Marriage Nationwide." Many in the LGBTQ+ community celebrated with tears of joy, feeling validated and recognized. Even some of the justices who dissented acknowledged the profound impact of the decision on American society. The ruling represented a major milestone in the fight for LGBTQ+ rights, symbolizing a shift toward greater acceptance and equality.

Figure 14: Our dialogue with GPT4 for generating corpus of varying levels of similarity for the original corpus from a CNN news.

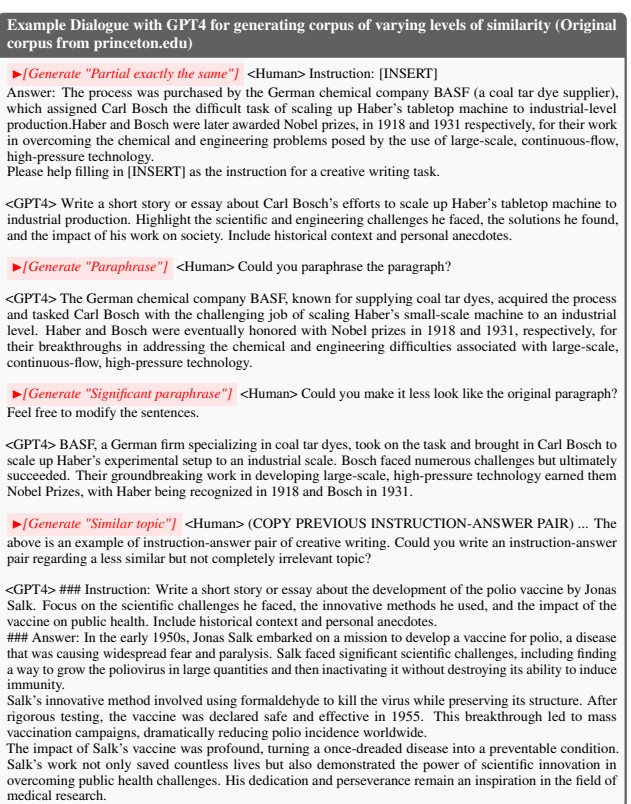

Figure 15: Our dialogue with GPT4 for generating corpus of varying levels of similarity for the original corpus from princeton.edu.

### E.5.1 ADDITIONAL EXPERIMENT: RELEVANT DOMAIN DETECTION

In this additional experiment, we evaluate the effectiveness of different data attribution techniques in identifying relevant domain-specific corpora within the Pile dataset. Specifically, we take a random batch of the validation corpus from sub-domains of the Pile dataset and evaluate the *normalized recall@1000 scores*. This metric measures the proportion of same-domain corpora as the validation corpora among the 1000 highest-valued corpora, normalized by the global proportion of this domain within the Pile dataset. As shown in Table 3, BM25 performs well on this task in general. Both the first-order and second-order In-Run Data Shapley methods outperform the influence function in detecting relevant domain corpora. **Analysis:** The influence function only uses information from the final trained models, which can result in highly noisy value scores since the removal of one training data point might have a negligible effect on the final model performance. In contrast, In-Run Data Shapley effectively leverages information from all intermediate checkpoints during model training, providing a more comprehensive and accurate assessment of each data point's value. While BM25 performs well on most of the domains, there are still cases where In-Run Data Shapley achieves better detection rate, which implies that gradient information may be more useful in determining semantic similarity to some extent.

|  | Github | EuroParl | ArXiv | PhilPapers | FreeLaw | HackerNews | StackExchange | Wikipedia (en) |
|---|---|---|---|---|---|---|---|---|
| **In-Run Data Shapley (1st order)** | 30.5 | 12.7 | 48.4 | 0.6 | 14.6 | 2.7 | 16.3 | 14.5 |
| **In-Run Data Shapley (2nd order)** | 29 | 14.8 | 50 | 0.4 | 16.9 | 3.8 | 20 | 18.9 |
| **Influence Function** | 17.9 | 4.1 | 41.7 | 0.5 | 5.7 | 2.6 | 6.5 | 5.5 |
| **BM25** | 18.8 | 6.3 | 44.8 | 1.1 | 72.6 | 9.9 | 37.2 | 18.6 |

|  | PubMed Abstracts | USPTO Backgrounds | PubMed Central | Enron Emails | NIH ExPorter | DM Mathematics | Ubuntu IRC | Gutenberg (PG-19) |
|---|---|---|---|---|---|---|---|---|
| **In-Run Data Shapley (1st order)** | 10.4 | 10.5 | 30 | 0.8 | 1.7 | 23.5 | 10.2 | 8.7 |
| **In-Run Data Shapley (2nd order)** | 16.6 | 11.1 | 31.1 | 1 | 1.9 | 24.5 | 11.5 | 15.3 |
| **Influence Function** | 10.2 | 7.4 | 26.9 | 0.2 | 1.2 | 10.1 | 4.3 | 8.8 |
| **BM25** | 28.6 | 23 | 53.7 | 1 | 4.1 | 34.6 | 12.5 | 25.9 |

Table 3: Results of relevant domain corpus detection experiment, where the metric is the normalized recall@1000 (i.e., the proportion of the same domain corpora as the validation corpora among the 1000 corpora with the highest values, normalized by the global proportion of this domain among the Pile dataset).

**Original Wikipedia Corpus**

In 2012, Radhi recruited new 'musicians' for OAG, who were selected from among the students of Akademi Seni Budaya dan Warisan Kebangsaan (). The new line-up consists of Qi Razali (drums/backing vocals - original drummer ...

**A corpus that is partially same**

### Instruction: Write a short story about Radhi and his band OAG as they recruit new members and blend traditional music with alternative sounds. Describe their creative challenges and successes leading up to their new releases. ### Answer: In 2012, Radhi recruited new 'musicians' for OAG, who were selected from among the students of Akademi Seni Budaya dan Warisan Kebangsaan (). The new line-up consists of Qi Razali (drums/backing vocals - original drummer ...

**Original Wikipedia Corpus**

In 2012, Radhi recruited new 'musicians' for OAG, who were selected from among the students of Akademi Seni Budaya dan Warisan Kebangsaan (). The new line-up consists of Qi Razali (drums/backing vocals - original drummer ...

**Synthetic "Similar topic" Corpus**

### Instruction: Write a short story about a classical violinist who decides to explore jazz music, detailing her first encounter with a jazz band. ### Answer: Elena, a classically trained violinist known for her precise and emotive performances ...

### E.6 COMPARISON WITH OTHER DATA ATTRIBUTION BASELINES ON SMALL-SCALE MODELS

In this section, we evaluate In-Run Data Shapley using standard benchmarks for data attribution techniques, including mislabeled data detection and data selection. While data attribution scores can generally indicate data quality, they are not specifically optimized for these benchmarks. Furthermore, approximating the Shapley value has intrinsic merit beyond these standard tasks (e.g., for royalty sharing of AI-generated contents (Wang et al., 2024a)). As such, this evaluation primarily serves as a *sanity check* to confirm that In-Run Data Shapley can reasonably reflect data values. However, state-of-the-art performance is not expected since the method is not designed specifically for these tasks.

We compare In-Run Data Shapley with several data attribution baselines, including Retraining-based Data Shapley (Ghorbani & Zou, 2019), KNN-Shapley (Jia et al., 2019a), influence function (Koh & Liang, 2017), Trak (Park et al., 2023), Empirical Influence Functions (Feldman & Zhang, 2020), and Datamodels (Ilyas et al., 2022). We also compare with LESS (Xia et al., 2024), an extension of TracIN-CP tailored for instruction tuning data selection.

As some baselines here require multiple model retrainings, we use a subset of 1,000 samples from CIFAR10 and ResNet18 architecture. We evaluate these techniques on two standard tasks for assessing data attribution methods: mislabeled data detection and data selection. We note that recent work (Wang et al., 2024d) questions the suitability of data attribution for data selection, but we include this metric due to its prevalence in existing literature.

**Experiment settings.** We use ImageNet-pretrained ResNet18 as the architecture in the experiment. We use Adam with a learning rate 0.001, weight decay of 1e-4, and label smoothing of 0.1 over 50 epochs. The learning rate is reduced by a factor of 0.1 every 10 epochs. The batch size is set to 64. For retraining-based techniques (Retraining-based Data Shapley, Empirical Influence Functions, Datamodels), we estimate the corresponding attribution scores with 1000 model training runs. For LESS, we use the checkpoints from epochs 10, 20, 30, 40, 50 and set the projection dimension to 2048 as recommended in the original paper. For Trak, we set the projection dimension to be 2048. For KNN-Shapley, we set K=5 and use the features extracted from the last linear layer of ResNet18. We also evaluate its performance by averaging the results using the last $T$ checkpoints where $T \in \{1, 5, 15, 25\}$.

**Mislabeled data detection.** The results in Table 4 show that KNN-Shapley achieves the highest performance in mislabeled data detection, likely due to its sensitivity to label changes. Retraining-based approaches (Retraining-based Data Shapley, Empirical Influence Functions, Datamodel) have the lowest performance, which can be attributed to Monte Carlo sample inefficiency and stochasticity during retraining, as noted in Wang & Jia (2023a). Among techniques requiring only one training run, Trak does not outperform others, aligning with observations in its original paper (Park et al., 2023) that ensembles are often necessary for high performance. LESS achieves better results by incorporating more information from intermediate checkpoints. In-Run Data Shapley and influence function achieve comparable performance, outperforming all other techniques except KNN-Shapley. The relatively high performance of In-Run Data Shapley is likely due to its deterministic estimation algorithm. Furthermore, it makes use of the information from all intermediate checkpoints.

**Data selection.** Table 5 shows that KNN-Shapley again excels when the selection budget is small. Interestingly, most data attribution techniques perform worse than random baseline for small selection budgets, aligning with observations that selecting data points solely based on attribution scores can harm dataset diversity (Wang et al., 2024d). For larger selection budgets (60%, 80%), single-training-run techniques achieve similar performance levels.

| | | |
|---|---|---|
| **Retraining-based Data Shapley** (Ghorbani & Zou, 2019) 
 0.582 (0.029) | **Empirical Influence Function** (Feldman & Zhang, 2020) 
 0.552 (0.017) | **Datamodels** (Ilyas et al., 2022) 
 0.52 (0.008) |
| **KNN-Shapley** (Jia et al., 2019a; Wang & Jia, 2023c) 
 **0.76 (0.018)** | **Influence function** (Koh & Liang, 2017) 
 0.654 (0.054) | **LESS** (Xia et al., 2024) 
 0.55 (0.032) |
| **1st Order In-Run Data Shapley (ours)** 
 **0.678 (0.045)** | **2nd Order In-Run Data Shapley (ours)** 
 **0.680 (0.048)** | **Trak (1 cpt)** (Park et al., 2023) 
 0.511 (0.012) |
| **Trak (5 cpts)** (Park et al., 2023) 
 0.542 (0.025) | **Trak (15 cpts)** (Park et al., 2023) 
 0.609 (0.028) | **Trak (25 cpts)** (Park et al., 2023) 
 0.617 (0.021) |

Table 4: AUROC scores of mislabeled data detection task with various data attribution techniques on CIFAR10 dataset. The higher the AUROC score is, the better the method is. The results are across three different training runs (the randomness comes from construction of corrupted datasets), where we show the standard deviation in (). Methods with top-3 performance are bolded.

| | 20% | 40% | 60% | 80% |
|---|---|---|---|---|
| Random | **0.350 (0.010)** | 0.461 (0.010) | 0.525 (0.004) | 0.559 (0.003) |
| Influence function (Koh & Liang, 2017) | 0.320 (0.033) | 0.450 (0.028) | 0.530 (0.015) | **0.580 (0.004)** |
| Trak (1 cpt) (Park et al., 2023) | 0.329 (0.021) | 0.443 (0.030) | 0.517 (0.016) | 0.572 (0.009) |
| Trak (5 cpts) (Park et al., 2023) | 0.325 (0.022) | 0.445 (0.031) | 0.519 (0.018) | 0.572 (0.009) |
| Trak (15 cpts) (Park et al., 2023) | 0.327 (0.028) | 0.446 (0.035) | 0.523 (0.018) | 0.573 (0.010) |
| Trak (25 cpts) (Park et al., 2023) | 0.332 (0.025) | 0.443 (0.035) | 0.520 (0.018) | 0.578 (0.015) |
| LESS (Xia et al., 2024) | 0.314 (0.012) | 0.460 (0.015) | **0.541 (0.006)** | **0.582 (0.006)** |
| Data Shapley (Ghorbani & Zou, 2019) | 0.317 (0.047) | 0.468 (0.010) | 0.527 (0.004) | 0.570 (0.008) |
| Datamodels (Ilyas et al., 2022) | 0.342 (0.004) | 0.465 (0.004) | 0.534 (0.010) | 0.559 (0.005) |
| Empirical Influence Function (Feldman & Zhang, 2020) | 0.342 (0.004) | 0.466 (0.016) | 0.530 (0.009) | 0.568 (0.010) |
| KNN-Shapley (Jia et al., 2019a) | **0.354 (0.017)** | **0.478 (0.007)** | 0.525 (0.015) | 0.563 (0.005) |
| 1st Order In-Run Data Shapley | **0.344 (0.009)** | **0.472 (0.003)** | **0.541 (0.006)** | **0.580 (0.004)** |
| 2nd Order In-Run Data Shapley | 0.342 (0.010) | **0.473 (0.008)** | **0.544 (0.005)** | **0.580 (0.004)** |

Table 5: Test accuracies when training ResNet18 on high-value data points selected by various data attribution techniques. To be able to compare with techniques that require model retraining, for each training run we randomly sample a size-1000 subset of CIFAR10 dataset (with 10% data points being mislabeled). The results are across three different training runs (the randomness comes from the construction of corrupted datasets), where we show the standard deviation in (). Methods with top-3 performance are bolded.

