# OpenReview forum: "Data Shapley in One Training Run"
_ICLR.cc/2025/Conference — ICLR 2025 Oral_

### Official Review · Reviewer_Vb7n · 2024-10-31

**Soundness:** 3
**Presentation:** 3
**Contribution:** 4
**Rating:** 6
**Confidence:** 4

**Summary:**

The paper introduces in-run Shapley, a data valuation method for SGD based models that estimates data values in one training run across different update steps.

The authors further provide enhanced ways of computing these data values with first and second order approximations.

The method is evaluated on the PILE dataset for GPT-2 and Pythia 4.10 against influence functions and outperforms them.

Remark: Thanks to the clarifications provided by the authors during rebuttal, I have raised Soundness from 2 to 3, Contribution from 3 to 4, and the overall score from 5 to 6.

**Strengths:**

The paper introduces an interesting and to me novel method to estimate data values in a single training run. This makes data valuation feasible to large scale data sets. Furthermore, the method is evaluated on such a data set.

**Weaknesses:**

The evaluation is performed against influence functions only. This way it is unclear whether the method only outperforms leave-one-out (approximated by influence functions) or is also better than the standard data Shapley value.

Other data valuation/pruning methods are dismissed due to scalability issues (Sec 5.3.1). However, to my understanding, both Memorization (Socher et al., 2022, Feldman and Zhang (2020)) and Supervised Prototypes (Socher et al., 2022) should be applicable to larger models and have been implemented on ImageNet scale by Socher et al. I'm uncertain if this generalizes to LLMS, but it should be discussed if not feasible.

It is specifically indicated that TRAK wouldn't scale to LLMs, yet according to my understanding, TRAK can function with checkpoints from a limited number of models (see Appendix "E.3 Proxies for model ensembles in compute-constrained settings" of the TRAK paper).
Given the evaluation of in-run Shapley on large datasets, I would elevate my vote if the authors either compare their method to the ones mentioned above or provide reasons why those methods don't scale to the dataset.

Minor Remarks
--------------------
OpenDataVal is a recent benchmark for data valuation. Although it is designed for small-scale data, it offers ready-to-use implementations of various algorithms and could be employed to benchmark In-Run Shapley against other methods. This would enhance the paper by demonstrating whether In-Run Shapley performs comparably or even superior to the standard Shapley value.
Some references are missing:
* Toneva et al. (2019) demonstrated that data points learned early in training and not forgotten in later epochs are less important to model performance than points that are frequently forgotten. The former can be omitted to reduce dataset size.. This reminds of the method introduced in the paper.
* Nguyen et al. (2023) proposed "A Bayesian Approach To Analysing Training Data Attribution In Deep Learning" to tackle the noise issue when training multiple models. This problem, referred to as "instability," is also mentioned in this paper but without citing Nguyen et al.

There are a few spelling mistakes especially in Sec. 5.3.3:
* In "data set," the word "set" is missing.
* The sentence "The results of this experiment have important implications for the current discussion on the copyright of generative AI." is repeated.

Sources
-----------
Remark: the reviewer is not the author of any of the papers below.
* Socher et. al (2022): Beyond neural scaling laws: beating power law scaling via data pruning
* Feldman and Zhang (2020): What neural networks memorize and why: Discovering the long tail via influence estimation
* Toneva et. al (2019): An Empirical Study of Example Forgetting during Deep Neural Network Learning
* OpenDataVal (2023): OpenDataVal: a Unified Benchmark for Data Valuation
* Nguyen et. al (2023): A Bayesian Approach To Analysing Training Data Attribution In Deep Learning

**Questions:**

I want to confirm whether I have understood the definition of the in-run Shapley value for a data point $z$. It is the “cumulative contribution of the data point $z$ across all gradient update iterations within a single training run”. Does this mean that in each gradient update (e.g., for each batch) the Shapley data value w.r.t. to this batch is computed and averaged at the end? This would mean that for each batch $b$ the marginal contribution of $z$ would be estimated on all possible subsets of $b$.

If so, I would be interested in an intuition how this value is related to the standard Shapley data value. This could be done conceptually but also by experimental evidence, e.g., by comparing the method to other visiting Shapley based estimates, as suggested above.

**Details Of Ethics Concerns:**

No ethical concerns.

---

> ### Author Response · Authors · 2024-11-23
>
> We thank the reviewer for the positive comments about our paper!
>
> **Q [Additional baseline comparisons]** *“The evaluation is performed against influence functions only ... both Memorization (Socher et al., 2022, Feldman and Zhang (2020)) and Supervised Prototypes (Socher et al., 2022) should be applicable to larger models ... It is specifically indicated that TRAK wouldn't scale to LLMs ...”*
>
> **A:** We thank the reviewer for the suggestion of having additional baseline comparisons, and actually this experiment was deferred to Appendix E.6 due to space constraints. In Appendix E.6, we evaluate In-Run Data Shapley on the standard tasks of mislabeled data detection and data selection, against a comprehensive set of baselines including *Retraining-based Data Shapley, KNN-Shapley, Influence Functions, TRAK, Empirical Influence Functions, Datamodels, and LESS*. The results demonstrate In-Run Data Shapley's competitive performance: In-Run Data Shapley's performance was among top-3 for almost all settings.
>
> For *“Memorization (Socher et al., 2022, Feldman and Zhang (2020)) and Supervised Prototypes (Socher et al., 2022) should be applicable to larger models and have been implemented on ImageNet scale by Socher et al.”*, we thank the reviewer for mentioning these two works.
>
> **Memorization Scores (Feldman & Zhang, 2020):** **(1) Methodological Distinction:** Memorization score is not a data attribution technique as it is independent of validation data, while the data attribution techniques aim to quantify the training data’s contribution to the model behavior on a validation point. The memorization score measures how much a model memorizes individual training points by computing the loss change on the training point itself when removed from training (we can think of it as replacing the validation data by the training data point itself).  **(2) The “influence score” proposed in (Feldman & Zhang, 2020) has already been compared with in Appendix E.6:** The same paper also proposes a “influence score” for data attribution (referred to as “Empirical Influence Functions” in Appendix E.6), which evaluates contribution to validation performance and has been included in our experiments in Appendix E.6. Like other retraining-based methods, Empirical Influence Functions **require multiple complete training runs** (1000 in our experiments), making it impractical for foundation models where a single training run can take weeks.
>
> **Supervised Prototypes (Socher et al., 2022):** **(1) Methodological Distinction:** Supervised Prototype is a technique for measuring *data point difficulty* through proximity to class distribution centers, independent of validation data. Similar to the Memorization score, this is not a data attribution technique for quantifying each training point's contribution to model performance on validation data. **(2) Robustness to Noise:** Although not a data attribution technique, we have additionally evaluated the performance of Supervised Prototypes on the data selection experiment in Appendix E.6. We want to stress that in our experiments, we inject noise by flipping labels for 10% of training data points to better simulate real-world scenarios with imperfect data quality. This setting differs from the original paper which assumes clean data (ImageNet is a very clean dataset for academic use). The results (shown in the last row) demonstrate that Supervised Prototypes perform poorly in such noisy settings. This is because it identifies mislabeled examples as "difficult" or "informative" cases that should be retained in the dataset. This limitation arises because Supervised Prototypes evaluate data points solely based on their relationship to class prototypes, without considering their actual impact on model performance. In other words, Supervised Prototype is **unable to differentiate between genuinely difficult examples (that could benefit learning) and harmful data points (like mislabeled examples that hurt model performance).** These data pruning methods are most effective on pruning pre-curated datasets where data quality issues have already been addressed, rather than raw datasets where distinguishing helpful from harmful data is crucial. In contrast, data attribution methods like ours leverage *validation set performance* to identify and assign negative values to harmful mislabeled data, making them more suitable for real-world scenarios where data quality cannot be guaranteed.
>
> *(Table copied from Appendix E.6 with other baselines' removed and Supervised Prototype added)*
>
> | Method | 20% | 40% | 60% | 80% |
> |--------|-----|-----|-----|-----|
> | Random | 0.350 (0.010) | 0.461 (0.010) | 0.525 (0.004) | 0.559 (0.003) |
> | 1st Order In-Run Data Shapley | 0.344 (0.009) | 0.472 (0.003) | 0.541 (0.006) | 0.580 (0.004) |
> | 2nd Order In-Run Data Shapley | 0.342 (0.010) | 0.473 (0.008) | 0.544 (0.005) | 0.580 (0.004) |
> | Supervised Prototype | 0.280 (0.028) | 0.390 (0.025) | 0.466 (0.033) | 0.516 (0.042) |

---

> > ### Comment · Reviewer_Vb7n · 2024-11-29
> >
> > Thank you for pointing to the results in the appendix of the papers. This clearifies parts of my concerns.

---

> ### Author Response · Authors · 2024-11-23
>
> (continued from the previous response)
>
> For *“It is specifically indicated that TRAK wouldn't scale to LLMs, ... I would elevate my vote if the authors either compare their method to the ones mentioned above or provide reasons why those methods don't scale to the dataset.”*, we thank the reviewer for mentioning TRAK paper.
>
> Thanks for the great question!
>
> While TRAK can indeed use checkpoints as suggested in their appendix, it actually faces significant scalability challenges when applied to foundation models. The main bottleneck is computational: **computing per-sample gradients for each checkpoint effectively incurs similar costs as training with batch size 1 for an entire epoch**. Since foundation models typically train for just one epoch, using multiple checkpoints becomes almost as expensive as performing multiple complete training runs. Additionally, TRAK's random projection step is memory-intensive for large models due to the gigantic projection matrix sizes.
>
> While TRAK was already in our comparisons in Appendix E.6, during the rebuttal period **we have additionally evaluated TRAK with checkpoint averaging** (Trak (5 cpts), Trak (15 cpts), Trak (25 cpts) are being newly added during the rebuttal period). While increasing the number of checkpoints does improve TRAK's performance, it remains significantly below In-Run Data Shapley, which leverages information from all training iterations. We also observe diminishing returns in TRAK's performance as more checkpoints are added, which is consistent with findings from the original paper.
>
> We sincerely thank the reviewer for the great question, which makes our experiments more comprehensive!

---

> > ### Comment · Reviewer_Vb7n · 2024-11-29
> > **Additional evaluation of TRAK**
> >
> > Thanks for putting some additional effort on evaluating TRAK with checkpoint averaging. This adresses one of my main concerns. As pointed out in my main review, one this point is adresses, I will re-evaluate the paper and raise the score.

---

> > > ### Author Response · Authors · 2024-11-29
> > >
> > > We greatly appreciate your careful review! Thanks so much for raising the score!
> > >
> > > We wonder whether we have addressed all your questions. As we mentioned in our rebuttal, the "memorization score" (Feldman and Zhang, 2020) and "Supervised Prototype" (Socher et al., 2022) were actually designed for different purposes than data attribution. In particular, they cannot effectively distinguish between difficult-to-learn data and bad data, which has been discussed in the literature [1] and shown in our additional comparison results in the rebuttal. We also want to highlight that Feldman and Zhang (2020) proposed a separate data attribution technique called the "influence score," which **we have already included in our comparison as "Empirical Influence Functions"** in Appendix E.6.
> > >
> > > Let us know if we have adequately addressed your questions! We're committed to ensuring our paper meets the high standards of the community.
> > >
> > > [1] Seedat, Nabeel, Fergus Imrie, and Mihaela van der Schaar. "Dissecting sample hardness: A fine-grained analysis of hardness characterization methods for data-centric AI." ICLR 2024

---

> ### Author Response · Authors · 2024-11-23
>
> **Q [Missing References]**
>
> **A:** We appreciate the reviewer highlighting relevant references.
>
> Thank you for bringing up Toneva et al. (2018). We have cited their work in our related works section. While their approach of tracking "forgotten" examples during training shares some temporal aspects with our method, their focus is on **measuring inherent example difficulty rather than data attribution** with respect to validation performance. We have clarified this key distinction in our discussion and note their method's limitations to classification tasks.
>
> We have cited Nguyen et al. (2023) in both our related works section and our discussion of Shapley value instability. While their work focuses on LOO-based data attribution methods (particularly influence functions) rather than Data Shapley's value instability, we agree it provides important context. We have included their work alongside other papers discussing influence function vulnerability (Basu et al. 2020, Søgaard et al. 2021) to provide a comprehensive view of stability challenges in data attribution methods.
>
> Thank you for suggesting OpenDataVal. We have added citations to OpenDataVal (Jiang et al., 2024) in our related works. I have also cited (Deng et al., 2024) which is another data attribution library focusing on influence function style data attribution techniques.
>
> Jiang, Kevin, et al. "Opendataval: a unified benchmark for data valuation." Advances in Neural Information Processing Systems 36 (2023).
>
> Deng, Junwei, et al. "dattri: A Library for Efficient Data Attribution." Advances in Neural Information Processing Systems 36 (2024).
>
> Toneva, Mariya, et al. "An Empirical Study of Example Forgetting during Deep Neural Network Learning." International Conference on Learning Representations. 2018.
>
> Basu, S., P. Pope, and S. Feizi. "Influence Functions in Deep Learning Are Fragile." International Conference on Learning Representations (ICLR). 2021.
>
> Søgaard, Anders. "Revisiting methods for finding influential examples." arXiv preprint arXiv:2111.04683 (2021).
> Nguyen, Elisa, Minjoon Seo, and Seong Joon Oh. "A Bayesian approach to analyzing training data attribution in deep learning." Advances in Neural Information Processing Systems 36 (2023).
>
> **Q [Typos]**
>
> **A** Thanks a lot for the catch! We have fixed the typo in the revised draft.

---

> > ### Author Response · Authors · 2024-11-23
> >
> > **Q [Comparison between In-Run Data Shapley and standard Shapley value]** *“I want to confirm whether I have understood the definition of the In-Run Shapley ... This would mean that for each batch $b$ the marginal contribution of $z$ would be estimated on all possible subsets of $b$.” “... how this value is related to the standard Shapley data value. This could be done conceptually but also by experimental evidence, e.g., by comparing the method to other visiting Shapley based estimates, as suggested above.”
> >
> > **A:** We appreciate the reviewer's valuable question. In-Run Data Shapley tracks each data point's contribution throughout the entire training process by accumulating its impact in every batch *where it appears*. Let us clarify two key aspects:
> > 1) **Definition**: In-Run Data Shapley is computed with respect to a *specific model training trajectory* with batch order $b_1, b_2, ..., b_T$. For each iteration $t$, we define a utility function $v_t$ that measures how much any subset $S$ of the training data would impact the model if used for that iteration's update. Importantly, if a data point $z$ is not selected in the current batch $b_t$, then adding $z$ to any subset $S$ doesn't change the utility (i.e., $v_t(S) = v_t(S \cup \{z\})$ for all $S$). This means that $z$'s Shapley value for that iteration ($\phi_z(v_t)$) must be zero. The final value for each data point will be the sum of its Shapley values across all iterations where it appears in the training batch.
> > 2) **Efficient computation**: Computing these values naively would require evaluating the marginal contribution of $z$ to every possible subset of its batch (or using random sampling). Our key technical contribution avoids this computational burden: first, by using Taylor approximations to derive closed-form expressions for the per-iteration Shapley values, and second, by developing "ghost" techniques that efficiently implement these computations without explicitly calculating individual gradients.
> >
> > For *“If so, I would be interested in an intuition how this value is related to the standard Shapley data value. This could be done conceptually but also by experimental evidence, e.g., by comparing the method to other visiting Shapley based estimates, as suggested above.”* From the high-level perspective, In-Run Data Shapley measures data points' contributions to a specific trained model, rather than their average contribution across multiple hypothetical training runs.
> >
> > **Concept comparison.** ​​Traditional Data Shapley faces a conceptual challenge with deep learning models: the training process is inherently random due to factors like mini-batch selection and initialization. However, it is unclear how to properly define and calculate Shapley values for randomized utility functions. While one could take the expectation over the learning randomness for utility function evaluations, this approach fails to capture how data contributions depend on crucial factors like initialization and training order. For instance, the same data point might be very valuable early in training but less important later [1]. In-Run Data Shapley, by encoding the specific sequence and randomness of a single training run into its utility function, provides a more fine-grained notion that reflects each data point's actual contribution to the deployed model.
> >
> > **Practical challenges of retraining-based Data Shapley.** There are also practical challenges with traditional Data Shapley. In machine learning, hyperparameters (like learning rate and batch size) typically need to be adjusted based on the size of the training dataset. This creates a **fundamental ambiguity** when computing Data Shapley values: when evaluating the utility $v(S)$ for different data subsets $S$, should we use the same hyperparameters as those optimized for the full dataset, or should we adjust them based on the subset size? This choice can significantly impact the calculated values. Furthermore, the Monte Carlo methods commonly used to estimate traditional Data Shapley values can **violate the fairness properties** that make Shapley values theoretically appealing in the first place.
> >
> > **Empirical Comparison.** On small-scale settings (Appendix E.6) where we can estimate retraining-based Data Shapley, we can see that In-Run Data Shapley significantly outperforms traditional approaches. This improved performance comes from producing more stable, reliable value estimates compared to the high variability seen in traditional Data Shapley calculations.
> >
> > [1] Paul, Mansheej, Surya Ganguli, and Gintare Karolina Dziugaite. "Deep learning on a data diet: Finding important examples early in training." Advances in neural information processing systems 34 (2021): 20596-20607.

---

> > > ### Author Response · Authors · 2024-11-25
> > >
> > > Dear Reviewer Vb7n,
> > >
> > > We sincerely thank you for your constructive feedback, which has helped us significantly improve our paper. Following your suggestions, we have made revisions including (1) new experiments with TRAK across different numbers of checkpoints (**Appendix E.6**), (2) an expanded discussion on comparing retraining-based and In-Run Data Shapley in **Appendix B.1 and B.2**, and (3) we have also carefully addressed all missing references. We truly value your expertise and would greatly appreciate any additional thoughts you might have on our work :)

---

### Official Review · Reviewer_ZCk2 · 2024-11-03

**Soundness:** 3
**Presentation:** 3
**Contribution:** 3
**Rating:** 8
**Confidence:** 3

**Summary:**

The paper presents a novel approach called In-Run Data Shapley that eliminates the need for model retraining to evaluate data contributions in machine learning. By leveraging gradient updates within a single training run, the authors develop a computationally efficient method suitable for large-scale models, such as foundation models like GPT-2. The work introduces efficient algorithms using Taylor expansions and techniques inspired by differential privacy to minimise runtime overhead. The study also demonstrates the practical utility of In-Run Data Shapley in data curation and understanding the impact of pretraining data.

**Strengths:**

* The introduction of In-Run Data Shapley is a significant advancement over existing retraining-based methods, offering a scalable solution to data attribution in large models.
* The case studies, particularly in data curation and copyright implications, highlight the real-world relevance of the proposed method.
* The closed-form derivations for the Shapley values using first- and second-order Taylor approximations are mathematically sound and well-explained.

**Weaknesses:**

* The method's requirement for validation data to be available before training might limit its applicability in scenarios where such data is not readily available.
* Even though the authors address memory concerns using gradient accumulation, the practicality of this approach in extremely large models with limited resources is not fully explored.
* The first- and second-order Taylor approximations are empirically validated, but there is limited discussion on how these approximations might affect the reliability of the data attribution scores, especially in more complex models.

**Questions:**

* The method requires validation data to be available before training. Can the authors elaborate on the potential limitations of this requirement and discuss scenarios where this may not be feasible? Are there alternative approaches that could make In-Run Data Shapley applicable when validation data is not pre-available?
* How does the use of first- and second-order Taylor approximations affect the reliability of the Shapley values in practice, especially in highly non-convex or complex loss landscapes? Can the authors provide additional empirical results or theoretical analysis to quantify these approximation errors in diverse settings?
* While the paper demonstrates efficiency improvements, how does the proposed method scale when applied to models even larger than GPT-2 or to models with different architectures, such as transformer-based models with sparse attention mechanisms? Can the authors provide insights or preliminary results on this?

---

> ### Author Response · Authors · 2024-11-23
>
> We thank the reviewer for the very positive assessment of our paper!
>
> **Q [Requirement of Validation Data]**
>
> **A:** Thank you for this thoughtful question about the validation data requirement. There are indeed several important scenarios where pre-available validation data could be challenging or infeasible: In online learning settings, where the data distribution may shift over time, pre-defined validation data might not effectively represent the current distribution. Similarly, in federated learning scenarios, privacy constraints might prevent access to validation data before training. Another challenging case is in few-shot or zero-shot learning tasks, where validation data from the target distribution is inherently scarce or unavailable.
>
> For these scenarios, we envision several potential alternative approaches:
> - First, as mentioned in Section 6, one could save model checkpoints during training and compute attribution scores once validation data becomes available. However, we acknowledge this approach's limitations - the choice of checkpoints can significantly impact performance, and storing multiple checkpoints for large models may be resource-intensive.
> - One direction might be to develop proxy metrics that don't require explicit validation data. For instance, one could potentially use training data statistics, gradient information, or model uncertainty measures to construct surrogate utility functions. These could provide meaningful attribution scores without requiring pre-defined validation data, though careful theoretical work would be needed to establish their properties and relationship to validation-based attribution.
> - Another possibility is to develop incremental updating schemes that can efficiently revise attribution scores as new validation data becomes available, rather than requiring all validation data upfront. This would be particularly valuable for online learning scenarios.
>
> We believe exploring these alternatives represents an important direction for future research that could significantly broaden the applicability of our approach.
>
> We agree that the requirement for validation data before training is a limitation of our current approach, and we explicitly acknowledged this in Section 6 of our paper. However, we would like to clarify that there are many practical scenarios where validation data is naturally available before training. These include working with public benchmark datasets that have established validation sets, participating in machine learning competitions where test metrics are known upfront, developing industrial applications where specific performance criteria are predetermined, and addressing regulatory requirements that specify validation metrics in advance. **These examples where validation data are available upfront are not edge cases—they represent core challenges in deploying trustworthy ML systems.** Our method provides an immediate and practical solution for these pressing use cases where data valuation can have a significant real-world impact.

---

> ### Author Response · Authors · 2024-11-23
>
> **Q [Computational efficiency for larger or different architectures?]** *"... the practicality of this approach in extremely large models with limited resources is not fully explored." "While the paper demonstrates efficiency improvements, ... models even larger than GPT-2 or to models with different architectures, such as transformer-based models with sparse attention mechanisms?"*
>
> **A:** The memory efficiency of our approach is particularly notable - we maintain constant memory overhead relative to regular training by never materializing per-sample gradient vectors. For computational efficiency, our first-order variant requires only one backpropagation pass, introducing minimal runtime overhead. During each gradient update iteration, while regular back-propagation has complexity $O(p)$, our ghost dot-product technique reduces the additional computation to $O(\sqrt{p})$ by separately computing dot products between activations and output derivatives rather than materializing full gradient vectors. This significant efficiency gain is achieved by leveraging information from regular training backpropagations, as detailed in Section 4.2.
>
> We demonstrate this efficiency empirically in Section 5.1, where the first-order variant achieves 92.5% of regular training throughput on GPT2. For larger models (larger $p$), the **relative** overhead should decrease since regular training cost grows as $O(p)$ while our additional computation grows more slowly as $O(\sqrt{p})$. This favorable scaling property suggests our method becomes more efficient for larger models.
>
> **Distributed Training Considerations:** However, we note that training large-scale models typically requires distributed computation across multiple GPUs. Adapting our method to such distributed training settings presents additional engineering challenges. For instance, efficiently computing gradient dot-products across model parallel shards requires careful asynchronous implementation to avoid blocking the regular training loop. While these challenges are primarily engineering in nature rather than theoretical limitations, due to our limited academic computing resources, we leave a thorough investigation of distributed implementations as important future work.
>
> **Architectural Flexibility:** Although this work focuses on GPT2 and Pythia, our "ghost" techniques naturally extend to transformer-based models as they primarily consist of linear layers (query/key/value projections). For specialized variants like sparse attention, the core computations still involve linear transformations, making our method applicable with appropriate technical adaptations. For other neural network architectures involving non-linear layers such as convolutions, similar gradient decomposition methods have been developed in literature [1], suggesting our approach can also be extended to these architectures.
>
> [1] Lee, Jaewoo, and Daniel Kifer. "Scaling up differentially private deep learning with fast per-example gradient clipping." PETS 2021

---

> ### Author Response · Authors · 2024-11-23
>
> **Q [Additional results for In-Run Data Shapley approximation error]** *“How does the use of first- and second-order Taylor approximations affect the reliability of the Shapley values in practice, especially in highly non-convex or complex loss landscapes?”*
>
> **A:** The accuracy of Taylor approximation is controlled by its remainder terms: $O(\eta^2)$ for first-order and $O(\eta^3)$ for second-order approximations, where $\eta$ is the learning rate. Since modern foundation model pretraining uses small learning rates (typically $\eta \leq 10^{-3}$), these remainder terms remain small, ensuring reliable approximations in non-convex landscapes.
>
> **Additional experiments on more learning rate choices.** To thoroughly evaluate the impact of learning rate on approximation quality, we have conducted additional experiments during the rebuttal time (**in Appendix E.2.1**) varying the learning rate from our original $3 \times 10^{-4}$ to $6 \times 10^{-4}$ and $5 \times 10^{-3}$ (notably, the latter is significantly larger than typical learning rates used in LLM pretraining). While the mean squared error (MSE) increases with larger learning rates as expected, the Spearman rank correlation between approximation and (Monte Carlo-estimated) groundtruth remains remarkably high ($\approx 0.99$ for $\eta = 6 \times 10^{-4}$). Even with the extremely large learning rate of $\eta = 5 \times 10^{-3}$, we still maintain a strong Spearman rank correlation of $\approx 0.8$. This preservation of rank order is particularly important since most applications of data valuation (e.g., detecting low-quality data) rely on relative rankings rather than absolute values.
>
> The practical effectiveness of our method is further validated through extensive experiments on downstream tasks detailed in Appendix E.6. Our method achieves comparable or better performance than most of the existing data attribution methods in both mislabeled data detection and data selection tasks. While these results do not directly compare the approximation with the groundtruth, they provide strong evidence that our approximation meaningfully captures data quality differences.
>
> We thank the reviewer for this insightful question. The additional experiments and analyses have been included in our revised manuscript.

---

> > ### Author Response · Authors · 2024-11-25
> >
> > Dear Reviewer ZCk2,
> >
> > Thanks again for your insightful comments and very positive assessment of our work! We have revised and incorporated additional experiments into our paper. We truly value your expertise and would greatly appreciate any additional thoughts/comments you might have on our work :)

---

> > > ### Comment · Reviewer_ZCk2 · 2024-11-26
> > > **maintain my score**
> > >
> > > I appreciate their efforts in clarifying the raised concerns, I am happy to maintain my score.

---

> > > > ### Author Response · Authors · 2024-11-27
> > > >
> > > > Thanks! We are grateful for your encouraging feedback on our work!

---

### Official Review · Reviewer_wuoD · 2024-11-03

**Soundness:** 3
**Presentation:** 3
**Contribution:** 3
**Rating:** 6
**Confidence:** 3

**Summary:**

This paper presents a new technique called "in-run data shapley" to eleminate model retraining for assessing data contributions. This seems a new angle to assess the data attribution and importance. The results shows that this can provide efficiency improvement, and help make more insights for data contribution. The paper also identifies and discussed potential implications for copyright in genAI and pretraining data curation.

**Strengths:**

1. The motivations, problem statement and objectives and contributions are provided clearly.

2. The mathematical induction and proof seem solid.

3. The results and implications of the new concept/techniques are described clearly.

**Weaknesses:**

Compared with the introduction, background, methods and mathematical induction, which are presented clearly, the results and analyses are a bit weak. It would be better to provide concrete examples with more deep analyses.

**Questions:**

1. Could you please provide an example on how your method used in-run data Shapley to calculate the data contribution? Is it possible to provide a figure? Even using a simple example data set would be very helpful. This cannot be replaced with a loss-time curve.

2. Is it possible to extend the method to regression and classification problems?

**Details Of Ethics Concerns:**

The current results do not seem to have major ethical issues, but it has ethical implications in copyright.

---

> ### Author Response · Authors · 2024-11-23
>
> We thank the reviewer for the positive assessment of our paper!
>
> **Q [Example of how to use In-Run Data Shapley to calculate data contribution?]** *“Could you please provide an example on how your method used in-run data Shapley to calculate the data contribution? Is it possible to provide a figure?”*
>
> **A:** Thank you for this excellent suggestion. We have created new figures to illustrate the two key aspects of In-Run Data Shapley:
>
> **(1) Tracking data contributions throughout the training process**: This [figure](https://ibb.co/SNw76Rr) shows the overall process of In-Run Data Shapley, which tracks data contributions throughout the training process. In each iteration, when a mini-batch of training data is selected (illustrated by the green/blue/brown arrows), we compute the contribution of each data point in that batch to the model's performance change for the single gradient update. These contributions are accumulated over time (like a “data value accountant”), as visualized by the growing bar chart at the bottom of each panel.
>
> From the algorithmic perspective, the key idea of In-Run Data Shapley is to **decompose a complete training process into individual gradient update steps**. As shown in this [figure](https://ibb.co/2WfSRJ6), instead of evaluating data contribution across the entire training process at once (top row), we compute the Shapley value for each gradient update iteration separately (bottom row) and then aggregate them.
>
> For each iteration $t$, we consider the "local utility function" $U^{(t)}$ that measures how much a subset of data points in batch $B_t$ contributes to reducing the validation loss in this specific update step. We calculate the Shapley value $\phi_z(U^{(t)})$ for each training point $z$ in the current batch $B_t$ with respect to this local utility function. For those $z \notin B_t$, their value will be 0 due to the **null player axiom** (see Section 3’s paragraph “Data Shapley for a single gradient update”).
>
> The final contribution score for each data point $z$ is then computed as the sum of its Shapley values across all iterations where it appears: $\phi_z = \sum_{t:z\in B_t} \phi_z(U^{(t)})$. This decomposition approach makes the computation tractable while maintaining the key properties of the Shapley value through the linearity axiom.
>
> **(2) Single-update Shapley Calculation (exact calculation)**: Within each iteration, we compute the Shapley value for each data point in the current mini-batch using a standard Shapley value calculation. The only thing here is that the utility function is defined as the model's performance change **in this single update step**. For example, consider a simple binary classification scenario with a validation point and a mini-batch of two training points $\{1, 2\}$. The utility function $U(S)$ for any subset $S$ of the mini-batch would be the change in validation loss when only using points in $S$ for this gradient update. Suppose we observe:
> - $U(\emptyset) = 0$ (no update)
> - $U(\{1\}) = 0.3$ (gradient update on point 1 only reduces validation loss by 0.3)
> - $U(\{2\}) = -0.2$ (gradient update on point 2 only increases validation loss by 0.2)
> - $U(\{1,2\}) = 0.4$ (gradient update on both points reduces validation loss by 0.4)
>
> The Shapley value for $z_1$ would be:
> $$\phi_{z_1} = \frac{1}{2}[U(\{1\}) - U(\emptyset)] + \frac{1}{2}[U(\{1,2\}) - U(\{2\})]$$
> $$= \frac{1}{2}(0.3 - 0) + \frac{1}{2}(0.4 - (-0.2)) = 0.35$$
>
> Similarly for $z_1$:
> $$\phi_{z_2} = \frac{1}{2}[U(\{2\}) - U(\emptyset)] + \frac{1}{2}[U(\{1,2\}) - U(\{1\})]$$
> $$= \frac{1}{2}(-0.2 - 0) + \frac{1}{2}(0.4 - 0.3) = -0.15$$
>
> **(3) Single-update Shapley Calculation (Taylor Approximation)**:
>
> However, computing $U(S)$ for all possible subsets becomes computationally intensive in practice. This motivates our development of efficient first-order and second-order approximations in Section 4, which allow us to compute these Shapley values using only gradient information without requiring actual model updates.
>
> Specifically, in Section 4 we introduce Taylor approximations of the utility function and use them to derive the first- and second-order In-Run Data Shapley. As an example, for first-order In-Run Data Shapley, we proved that it is equal to the gradient dot product between the training and validation data. The [figure](https://ibb.co/X5YPXt2) illustrates the *first-order* In-Run Data Shapley values in a single iteration. Intuitively, if a training point's gradient aligns well with the validation gradient (like $z_1$ in the figure), it receives a positive value since updating the model in this direction would reduce the validation loss. Conversely, if the gradients point in opposite directions (like $z_2$), the point receives a negative value since it would increase validation loss.
>
> Our paper further develops efficient "ghost" techniques to compute these gradient operations without materializing individual gradients, making the method highly practical for large-scale models.

---

> > ### Author Response · Authors · 2024-11-23
> >
> > **Q [In-Run Data Shapley for Regression and Classification problem?]**
> >
> > **A:** Yes, In-Run Data Shapley can be directly applied to regression and classification problems, as our method is agnostic to the specific learning task. The key requirements for our method, aligning naturally with standard practices in modern machine learning, are: (1) The model is trained using batch SGD optimization. (2) We can compute gradients of the loss function with respect to model parameters. (3) We have a validation set to measure model performance.
> >
> > The formulation in our paper naturally accommodates different loss functions. For regression, we could use MSE or MAE loss. For classification, we could use cross-entropy loss or other classification-specific losses. **In Appendix E.6, we have additional experiments on classification tasks** (on CIFAR10) where the loss function is cross-entropy loss. In the experiments in the main paper, since we mainly focus on the language model pretraining, we use the standard next-token prediction loss.
> >
> > Our theoretical framework for approximating utility functions via Taylor expansion and the 'ghost' techniques for efficient computation remain valid regardless of the specific loss function, as long as the loss function is differentiable. The only adjustment needed would be to use the appropriate task-specific loss function. Overall, In-Run Data Shapley framework offers broad applicability across machine learning tasks, extending well beyond the specific experiments presented in our paper.

---

> > > ### Author Response · Authors · 2024-11-25
> > >
> > > Dear Reviewer wuoD,
> > >
> > > We greatly appreciate your valuable suggestions regarding the visual presentation of our methodology. We hope you find the new figures and examples we provided in our rebuttal intuitive and clear. Based on your feedback, we have also updated our manuscript to enhance overall clarity. We would be grateful to hear if you have any additional thoughts on our work :)

---

### Official Review · Reviewer_roW8 · 2024-11-09

**Soundness:** 4
**Presentation:** 4
**Contribution:** 4
**Rating:** 10
**Confidence:** 3

**Summary:**

This paper introduces In-Run Data Shapley, a novel principled way of attributing the data contribution for deep learning models within one training run. A classical Data Shapley technique requires retraining which is computationally prohibitive in practice for large models, and more importantly, it quantifies data contribution for a general learning algorithm, and not for a particular model at hand. Related to this, data contribution for a single model is of crucial importance for properly investigating fairness and copyright implications.

In-Run Data Shapley is powered by the fact that model training is performed iteratively, and Shapley values can be derived analytically for each model update step via gradient dot products or gradient-Hessian-gradient products between training and validation data. The authors propose an efficient method to approximate In-Run Data Shapley values during one training run via first- and second-order Taylor expansions, and compute these values in one and two backpropagation, respectively, inspired by the ghost clipping technique. This incurs negligible computational overhead, making In-Run Data Shapley highly usable in practice. Furthermore, the authors empirically evaluate their approach on Pile dataset, highlighting several important insights re: data curation and cleanliness of the dataset, dynamic data contribution throughout the entire training process, and the degree of data contribution in copyright issues.

**Strengths:**

The proposed method is novel, efficient and promising. The presentation of the paper is very clear and succinct; technical contributions are clearly stated and rigorously developed. The authors carefully position their work and acknowledge several limitations which they claim will be a part of future work. Empirical evaluation is thorough and the paper raises important observations re: societal implications of the large model training.

**Weaknesses:**

I do not see any major weaknesses of the paper.
Minor (typo): line 500 - sentence "The results of this experiment..." appears twice in a row

**Questions:**

I would like to know the authors' opinion of applying the general idea of In-Run Data Shapley to other use-cases in machine learning, such as hyperparameter importance during the HPO run. More generally, do the authors envision transferring this idea to other domains, especially in the case where gradient information is not available?

---

> ### Author Response · Authors · 2024-11-23
>
> We thank the reviewer for the very positive comments!
>
> **Q** *“I would like to know the authors' opinion of applying the general idea of In-Run Data Shapley to other use-cases in machine learning, such as hyperparameter importance during the HPO run”*
>
> **A**
> Thank you for this insightful question. We believe the core idea of In-Run Data Shapley—decomposing contribution analysis into per-iteration assessments—could be extended beyond data attribution to other applications in machine learning.
>
> **Hyperparameter Importance.** ​​The framework could potentially be adapted to evaluate hyperparameter contributions during training. One possibility is to set a "baseline" hyperparameter value and assess how choosing a different value impacts each training iteration compared to this baseline. For instance, when evaluating learning rate choices, we could measure how using a specific learning rate value affects model updates compared to using a baseline learning rate. For differentiable hyperparameters, we could leverage Taylor expansion to approximate this difference; for non-differentiable ones, zero-order methods could potentially be used. By accumulating these contributions across training iterations, we could understand hyperparameter impact without requiring multiple complete training runs. At a high level, **this view unifies our treatment of training data and hyperparameters** - both can be seen as choices that influence each training iteration, where we aim to quantify their impact against baseline scenarios. While technical challenges remain in adapting our method, this direction presents an interesting opportunity for future research.
>
> **Feature attribution (e.g., context-attribution for language models).** Another possible extension we envision is feature attribution for machine learning models. Feature attribution aims at understanding how each part of the input (like individual words in a sentence or pixels in an image) influences a model's final prediction. When a model makes a prediction, the input features are processed through multiple layers, with each layer transforming the information before passing it to the next layer. Feature attribution aims to track how this information flows and transforms through the network to determine each input feature's contribution to the final output. We envision a **unified theoretical framework** connecting training data attribution and feature attribution, drawing parallels between how information flows **during training (through gradient updates)** and **during prediction (through layer operations)**. This could lead to more efficient methods for explaining model behavior, particularly for complex architectures like transformers.
>
>
>
>
>
>
> **Q** *“Do the authors envision transferring this idea to other domains, especially in the case where gradient information is not available?”*
>
> **A** Thank you for this interesting question!
>
> **(1)** Consider scenarios where **models are trained with gradient descent** but the **data attribution algorithm only has black-box access to intermediate checkpoints** (e.g., when a third party wants to audit data influence during training). Here, zero-order methods could be used to estimate the impact of each training point by evaluating model behavior with and without that point in each iteration, without requiring access to gradients. While this approach may be computationally more expensive, it leverages the key insight of analyzing contributions iteratively rather than requiring complete retraining, and the technical adaptations can be interesting future works.
>
> **(2)** Our framework could potentially extend to **iterative learning algorithms that don't use gradient descent**, such as k-means clustering or decision tree learning. Though these algorithms update models differently, they still proceed iteratively (e.g., k-means alternates between assignment and update steps; decision trees are built through recursive partitioning). For such algorithms, we could analyze how each data point influences these discrete update steps and aggregate these influences across iterations, similar to our approach with gradient-based training.
>
> More broadly, we believe that the fundamental principle introduced in this work extends far beyond data attribution. As discussed in our previous response about feature attribution, this framework could offer new perspectives for understanding any iterative process where we want to quantify the impact of different components. We are excited about the potential applications of these ideas across various domains in machine learning and beyond. Thanks again for the great question!
>
>
> **Q [Typos]**
>
> **A** Thanks a lot for the catch! We have fixed the typo in the revised draft.

---

> > ### Author Response · Authors · 2024-11-25
> >
> > Dear Reviewer roW8,
> >
> > Thank you for your very positive assessment and valuable questions about the possibility of extending In-Run Data Shapley's core idea to other fields. We have incorporated our discussion in Appendix B.3. We truly value your expertise and would greatly appreciate any additional thoughts you might have on our work :)

---

### Author Response · Authors · 2024-11-25

We thank all reviewers for their thoughtful feedback and constructive suggestions. We are glad that our work received very positive feedback! We have carefully read the reviews and made substantial modifications to strengthen the paper. All modifications are highlighted in blue. Here's a summary of our major revisions:

- **Additional discussion on the potential of the general idea of In-Run Data Shapley (Reviewer roW8).** We have incorporated our discussion on the possible directions of generalizing the core idea of In-Run Data Shapley in **Appendix B.3**.

- **Additional figure on the core idea and algorithm overview of In-Run Data Shapley (Reviewer wuoD).** We have incorporated an additional Figure in **Appendix B** to better illustrate the core idea and algorithm of In-Run Data Shapley.

- **Additional experiments on Taylor approximation errors under various learning rates (Reviewer ZCk2).** We have conducted additional experiments during the rebuttal period in **Appendix E.2.1** varying the learning rate to systematically evaluate and demonstrate the robustness of our algorithm.

- **Additional baseline comparisons (Reviewer Vb7n).** We have conducted new experiments evaluating TRAK with different numbers of checkpoints. We have also revised the main text to highlight our comprehensive experiments comparing against a wide range of established baselines on standard benchmarks of mislabeled data detection and data selection in **Appendix E.6**.

- **Discussion on the comparison between retraining-based and In-Run Data Shapley (Reviewer Vb7n).** We have revised and enriched our discussion on the comparison between Retraining-based and In-Run Data Shapley in **Appendix B.1 and B.2**.

- **Typos & Missing References (Reviewer roW8, Vb7n).** We have fixed all of the them. Thanks so much for the catch!

As the discussion period is coming to an end, if there are any additional questions/comments about our work, we are more than happy to address them!

---

### Public Comment · ~Yaoxin_Ge1 · 2025-04-11
**Question on utility decomposition.**

The main idea of the In-Run Data Shapley is $U(S)=\sum_t U^{(t)}(S)$, but I found this decomposition is wrong following the construction of $U^{(t)}(S)$. Consider there are two data points $\\{A,B\\}$, and the model is updated with $A$ in the first iteration and with $B$ in the second iteration, $U(S)$ is the utility function introduced in Section 2.1.
Then, in the first iteraion:

$U^{(1)}(\\{A\\})=\text{The performance of using \\{A\\} to train}-0=U(\\{A\\})$,

$U^{(1)}(\\{B\\})=0$,

$U^{(1)}(\\{A,B\\})=\text{The performance of using} \\{A\\}\cap \\{A,B\\} \text{ to train}-0=U(\\{A\\})$,

and, in the  second iteration:

$U^{(2)}(\\{A\\})=0$,

$U^{(2)}(\\{B\\})=\text{The performance of using \\{B\\} to update}-\text{The performance of using \\{A\\} to train}=U(\\{A,B\\})-U(\\{A\\})$,

$U^{(2)}(\\{A,B\\})=U(\\{A,B\\})-U(\\{A\\})$,

so $U^{(2)}(B)+U^{(1)}(B)\neq U(B)$. That means the linear combination does not match the utility function, and in this example, the In-Run Data Shapley is just calculating the marginal contribution for each data point. Did I miss or misunderstand something?

---

> ### Public Comment · ~Jiachen_T._Wang1 · 2025-04-12
>
> Thank you for your interest. You are correct that if $U(S)$ represented the utility of the *entire end-to-end training process* in the traditional sense, the decomposition $U(S) = \sum_t U^{(t)}(S)$ would of course not true in general. That's not what this paper is proposing. The core point is that **In-Run Data Shapley defines its "global" utility function $U(S)$ differently** than the traditional Retraining-based Data Shapley.
>
> In the original Data Shapley literature, $U(S)$ typically defines utility as the final performance (e.g., accuracy, loss) of a model trained on a data subset $S$. However, things become more complicated in the context of deep learning. A specific deep learning training run is an *iterative process*. Defining utility $U(S)$ for a specific run makes it a *sequence function* rather than the pure *set function* required by the standard Shapley value framework. Extending Shapley axioms to sequence functions is not straightforward. The previous works often define $U(S)$ as "expected model utility across all possible training runs" instead to circumvent this conceptual issue. One contribution of this paper is initiating the study of "model-specific data attribution". This is important for applications such as training diagnosis or model decision interpretation, where you care about a specific training run instead of the expected model utility across all possible training runs.
>
> Our paper provides a novel approach by decomposing the training process into individual iterations, where each "local utility function $U^{(t)}$" is a set function, allowing efficient Shapley value approximation. We then define the "global utility function" as $U(S) := \sum_t U^{(t)}(S)$. In other words, our $U(S)$ has a different definition compared to the original. We interpret In-Run Data Shapley as "the cumulative data contribution throughout the training run." From a game theory perspective, this approach can be regarded as a special case of "asymmetric Shapley value" (https://arxiv.org/abs/2411.00388). It also closely relates to "instrumental value" (https://arxiv.org/pdf/2412.18140) which considers context-dependent valuation.
>
> We hope this makes sense. Please feel free to ask if there are any remaining questions. In a follow-up [work](https://openreview.net/forum?id=uHLgDEgiS5&noteId=qOhyOKUxlt) (also published at ICLR'25), we did some further studies on this "model-specific data attribution" problem, in case you are interested.
>
> PS: We appreciate your thoughtful inquiry. However, we believe the title 'Is there a theoretical mistake in this paper?' may set an unnecessarily critical tone that could influence other readers' perceptions, especially if they do not have enough time to read. We kindly suggest revising the title to a more neutral framing such as 'Question on utility decomposition' or "Question on theoretical foundations". This would better reflect the constructive nature of academic exchange.

---

> > ### Public Comment · ~Yaoxin_Ge1 · 2025-04-14
> >
> > Thank you for your reply, I think now I have no more questiongs. And sorry for the title, I've just realized that it may cause trouble and I've already revised it.

---

### Public Comment · ~Allen_Schmaltz1 · 2025-06-11
**"Instance/prediction-specific" data attribution methods**

An older approach for data attribution with the models with non-identifiable parameters (e.g., LLMs) is to make a semi-supervised connection to the observed data, conditional on the output prediction, by adding a bottleneck ("exemplar") layer to the model, and re-casting the prediction as a function over the training set's labels and representation-space via a metric-learner approximation. How do we know that the matched exemplars are actually relevant, or equivalently, that the approximation is faithful to the original model? One simple (but meaningful) metric is whether the prediction of the metric-learner approximation matches the class of the prediction of the original model, and if they do not, the discrepancies should be concentrated in low probability regions. Remarkably, relatively simple functions over the representation space and labels achieve that behavior. (In this way, such metric-learner approximations are a fundamentally different concept than Shapley-based metric-learner approaches, and provide an alternative approach for addressing the questions in the final sentence of Appendix B.1.1. Since In-Run Data Shapley requires validation sets, such alternative approaches are applicable in the same settings.) This line of work was introduced in the following paper, which appeared in the journal Computational Linguistics and was presented at EMNLP 2021: "Detecting Local Insights from Global Labels: Supervised & Zero-Shot Sequence Labeling via a Convolutional Decomposition" https://doi.org/10.1162/coli_a_00416.

Note that these older methods also address feature attribution, a proposed future extension of In-Run Data Shapley mentioned in Appendix B.3. When projection to a lower resolution than the available labels is desired for data analysis (e.g., from the document level to the word level for error detection), the aforementioned method provides a straightforward means of defining the applicable inductive bias via the loss (e.g., encouraging one feature detection per document, or multi-label detections, as shown in subsequent work) over the exemplar layer decomposition. In other words, we can decompose an LLM's document-level prediction to word-level feature detections, and then map to the labeled data in the support set. This works with models of arbitrary scale; the dense matching via the distilled representations from the filter applications of the exemplar layer requires computation on the order of commonly used dense retrieval mechanisms, and the other transforms require minimal additional overhead.

The punchline is that the key advantage relative to alternative data-attribution methods is this then leads to methods with which we can close the loop on the connection between the data, the representation space, [the features,] the predictions, and the predictive uncertainty at the instance level. See my works from 2022 and more recently for further details (e.g., https://arxiv.org/abs/2502.20167), including incorporating uncertainty-aware verification and interpretability-by-exemplar as intrinsic properties of the LLMs, themselves (i.e., for both train- and test-time search/compute/generation), rather than as a post-hoc approximation.

I would argue that data attribution tied to estimators of the predictive uncertainty is more principled than achieving the Shapley properties 1-3 described on page 1 (even if they were exactly achieved, rather than via an approximation), since it directly translates to the quantities of interest needed when using LLMs in practice. It also then becomes possible to estimate class-conditional contributions and to avoid marginalizing over distinctions across high/low-uncertainty partitions and high/low-variance partitions of the data, with concomitant benefits for constraining the output, or e.g., making targeted expansions of the support for a given instance. That is, in practice, we typically seek data attribution that is instance/prediction-specific, rather than algorithm-specific or model-specific (using the terminology of this paper).

---

### Meta-Review · Area_Chair_pnnD · 2024-12-20

**Metareview:**

The paper presents an approach aimed to evaluate the contributions of the data samples to the model.
This goal (attributing a relevance score to data samples w.r.t. the eventual model) is at the core of scientific, technical and legal concerns (privacy, copyright).
The method is based on the accumulation of Shapley value at each gradient iteration, with affordable memory and computational extra requirements.
One limitation is the availability at training time of validation samples.

**Additional Comments On Reviewer Discussion:**

Reviewers and area chair liked this paper; its extensions (e.g. regarding the impact of the hyper-parameter values, subject to differentiability) are also very interesting. The authors did an excellent job in their rebuttals, in particular clarifying the scope of the approach compared to the state of the art.

---

### Decision · Program_Chairs · 2025-01-22

Accept (Oral)